# Estrogen receptor beta in astrocytes modulates cognitive function in mid-age female mice

Noriko Itoh [1], Yuichiro Itoh [1], Cassandra E. Meyer[2], Timothy Takazo Suen[1], Diego Cortez-Delgado[1], Michelle Rivera Lomeli[1], Sophia Wendin [1], Sri Sanjana Somepalli[1], Lisa C. Golden[1], Allan MacKenzie-Graham[1,2] & Rhonda R. Voskuhl [1] ✉

Menopause is associated with cognitive deficits and brain atrophy, but the brain region and cell-specific mechanisms are not fully understood. Here, we identify a sex hormone by age interaction whereby loss of ovarian hormones in female mice at midlife, but not young age, induced hippocampal-dependent cognitive impairment, dorsal hippocampal atrophy, and astrocyte and microglia activation with synaptic loss. Selective deletion of estrogen receptor beta (ERβ) in astrocytes, but not neurons, in gonadally intact female mice induced the same brain effects. RNA sequencing and pathway analyses of gene expression in hippocampal astrocytes from midlife female astrocyte-ERβ conditional knock out (cKO) mice revealed Gluconeogenesis I and Glycolysis I as the most differentially expressed pathways. *Enolase 1* gene expression was increased in hippocampi from both astrocyte-ERβ cKO female mice at midlife and from postmenopausal women. Gain of function studies showed that ERβ ligand treatment of midlife female mice reversed dorsal hippocampal neuropathology.

As advances in healthcare lead to longer lifespans, cognitive decline associated with brain aging has become a major concern. Brain aging is associated with brain atrophy and neurodegeneration. It is also a risk factor for susceptibility and progression in many neurodegenerative diseases[1]. A better understanding of the effect of brain aging during health can provide insights into the effect of brain aging during disease.

While some factors of aging may not be reversible, others may be, such as the loss of endogenous sex hormones with aging. Menopause and andropause are each associated with cognitive decline in healthy people and are thought to worsen disabilities in Alzheimer's disease (AD) and multiple sclerosis (MS)[2–4]. Since testosterone is converted to estrogen in brain by aromatase, a decline in either sex hormone would result in less ligation of estrogen receptors in brain. This is relevant to neurodegeneration during aging since estrogens have shown

neuroprotective properties when optimized for estrogen type and dose as well as timing of administration and age of recipients[5–8].

The study of sex differences in health and disease is a way for basic research to be grounded in known clinical observations. Sex differences in disease can be mechanistically disentangled at the laboratory bench, with findings translatable to the clinic as treatment trials optimally tailored for women or men[9–11]. With regard to neurodegeneration, menopause in healthy women involves cognitive difficulties, termed "brain fog", that affect hippocampal-dependent verbal memory, attention, and processing speed as quantified by objective cognitive testing[12,13]. Also, healthy women have worse brain gray matter atrophy as assessed by magnetic resonance imaging (MRI) after age 65 years[14,15]. Alzheimer's disease (AD) is more common in women over age 60, which is not accounted for merely by greater longevity[16], and the rate of progression from mild cognitive impairment (MCI) to AD

[1]Department of Neurology, David Geffen School of Medicine at UCLA, Los Angeles, CA, USA. [2]Ahmanson-Lovelace Brain Mapping Center, Department of Neurology, David Geffen School of Medicine at UCLA, Los Angeles, CA, USA. ✉e-mail: rvoskuhl@mednet.ucla.edu

appears higher in women at older ages[16]. In multiple sclerosis (MS), older women have worsening of disabilities and more gray matter atrophy after menopause[17–19]. In contrast, at younger ages, namely before age 65 years, healthy men compared to women appear to have worse gray matter atrophy[20–22]. Also, at these relatively younger ages, men may be at greater risk for MCI[23], and the rate of progression from MCI to AD has been reported to be higher in men[16]. So too in MS, gray matter atrophy and cognitive deficits are worse in relatively younger men at mean ages in their early 40s[24–27]. Thus, differential timing of menopause versus andropause may impact neurodegeneration with brain aging[1]. Andropause in men starts at age 30 years with gradual decline of testosterone to age 70 years. In contrast, sex hormones are maintained in women through midlife until menopause at mean age 52 years with an abrupt loss of estradiol and progesterone. A neuroprotective effect of sex hormones is consistent with men demonstrating more risk for neurodegeneration before age 50 years, while women are more at risk after age 55 years. These sex differences in brain aging in humans serve as background and rationale for doing mechanistic studies addressing sex differences in brain aging in mice.

A sex-specific approach can provide mechanistic insights into age related cognitive decline. Sex differences in healthy brain structure and function are well established and exist across species[28,29], indicating a biologic effect of either sex hormones or sex chromosomes[9]. A brain region-specific approach is required since there are known differences in gene expression from one brain region to another within astrocytes[30–32], microglia[33], neurons[34], and oligodendrocytes[35]. Finally, a cell-specific and receptor-specific approach is required since effects of estrogen receptors (ERα and ERβ) can be either synergistic or antagonistic depending on cell type[36,37].

Sex differences in cognitive aging have been previously described in animal models[38]. Age related estrogen loss impairs cognition and is thought related to neurodegenerative changes in the hippocampus and prefrontal cortex[39]. Estrogen receptor beta (ERβ) ligand treatment has induced beneficial effects on recognition memory in ovariectomized rats using object recognition and placement memory tasks with effects on monoaminergic systems in hippocampus and prefrontal cortex[40]. Also, treatment with selective ERβ agonists reduced hot flash-like symptoms and enhanced spatial and object recognition memories in young ovariectomized mice[41]. Neuron-glia signaling contributes to synaptic changes and cognitive impairment during aging and neurodegenerative diseases[42–47]. Whether ERβ activation in neurons or glia can modulate cognitive decline during aging remains unclear.

Here, ligation of ERβ in astrocytes, but not neurons, is shown to mediate protection from hippocampal-dependent cognitive decline, dorsal hippocampal atrophy by in vivo MRI, and dorsal hippocampal neuropathology in female mice at midlife, but not young age. Further, genome wide analyses of gene expression in hippocampal astrocytes from midlife female mice with selective deletion of ERβ in astrocytes revealed dysregulation of genes involved in glucose metabolism. Lastly, ERβ ligand treatment of midlife female mice mitigated cognitive deficits and reversed dorsal hippocampal neuropathology.

## Results

### Sex differences occur in brain substructure atrophy during aging, with female mice showing relative protection at midlife followed by abrupt volume loss thereafter

First, we determined if a biomarker of brain aging in humans, namely substructure atrophy, might be sensitive in detecting neurodegeneration with aging in mice. We collected in vivo MR images in both female and male C57BL/6 mice at three ages: young (3–4 months), midlife (12–14 months), and old (20–22 months). See overview schematic of mice at various ages, either gonadectomized (GDX) or sham-treated (Fig. 1a). Atlas-based morphometry was used to analyze substructure volumes (Fig. 1b). Sham-treated females and males each

showed significant atrophy at old age as compared to young age in frontal cortex and striatum (Fig. 1c, d). However, at midlife there was a sex difference, with females showing relative protection as compared to males. The trajectory in males (blue) was gradual atrophy from young to midlife to old ages. In contrast, females (red) showed no atrophy from young to midlife, but thereafter had an abrupt drop in substructure volumes from midlife to old age. Further, the dorsal hippocampus is primarily involved in cognition and memory and is analogous to the posterior hippocampus in humans which is known to atrophy with age[48,49]. In dorsal hippocampus, sham-treated female mice had no atrophy from young to midlife, but had significant atrophy at old age (Fig. 1e). Whole hippocampus also showed atrophy from midlife to old age in sham-treated female, but not male, mice. (Supplementary Fig. 1). Thus, female mice were relatively protected against aging associated region-specific brain atrophy at midlife, but thereafter the trajectory of atrophy was striking.

### The effect of loss of endogenous ovarian hormones on dorsal hippocampal atrophy as measured by in vivo MRI

Given the sex differences observed in regional brain atrophy by in vivo MRI at midlife in sham-treated female mice, we next determined the effect of loss of endogenous sex hormones on atrophy of the hippocampus and its substructures in females at midlife. GDX versus sham surgery occurred at 2 months of age (see schematic, Fig. 1a). The effect of loss of ovarian hormones on hippocampal volumes at midlife was compared with hippocampal volumes at old age by collecting in vivo MR images from GDX and sham female mice at midlife and old age in a separate, independent experiment (Fig. 1f–h). In dorsal hippocampus, GDX females at midlife had smaller dorsal hippocampal volumes compared to sham females at midlife, with volumes in GDX midlife similar to those observed at old age (GDX and sham) (Fig. 1g). Whole hippocampus (Fig. 1f) showed trends similar to dorsal hippocampus, while ventral hippocampus (Fig. 1h) showed no effect, revealing hippocampal subregion-specific effects. In contrast to effects observed at midlife, GDX did not induce atrophy of dorsal hippocampus in female mice at young age (Supplementary Fig. 2).

### Loss of endogenous ovarian hormones induces hippocampal-dependent cognitive impairment in female mice at midlife

Next, we assessed how aging affects spatial reference memory using Morris Water Maze (MWM) testing at young, midlife, and old ages (see schematic, Fig. 1a). Consistent with previous observations in wild-type controls in mouse models of Alzheimer's disease, there was no significant impairment of spatial memory in healthy, sham-treated, C57BL/6 wild type female or male mice with aging (Fig. 2a, b). When we determined the effect of removal of endogenous ovarian hormones on cognitive behavioral testing in females and males at midlife, we found a sex difference. GDX female mice showed impairment of spatial reference memory at midlife, while GDX males performed well (Fig. 2c). Next, an interaction between loss of ovarian hormones and aging was discovered. GDX females at midlife had cognitive impairment (Fig. 2c), but neither GDX females that were young (Fig. 2c), nor sham-treated females that were midlife (Fig. 2a), had cognitive impairment. Together, this revealed a sex hormone by age interaction, whereby spatial memory impairment was due to both loss of ovarian hormones and aging.

To further investigate hippocampal-dependent cognitive impairment induced by loss of endogenous ovarian hormones in female mice at midlife, we performed Y maze testing to assess working memory. There was significant working memory impairment in GDX females at midlife compared to sham at midlife, as well as compared to young females (GDX and sham), (Fig. 2d). This result extended the observation of a sex hormone by age interaction from spatial reference memory to working memory. Together, two hippocampal-dependent behavioral tasks demonstrated that loss of endogenous ovarian

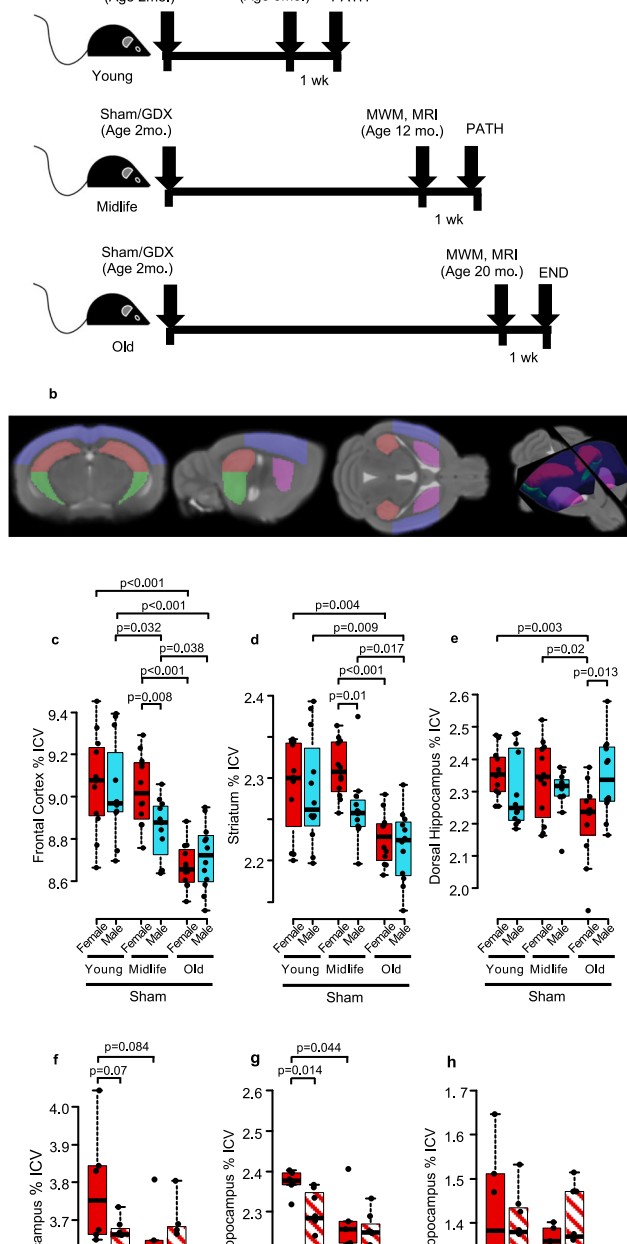

**Fig. 1 | Female mice show abrupt brain substructure volume loss after midlife compared to males, and ovariectomized females have smaller dorsal hippocampal volumes than sham treated females at midlife. a** Schematic showing timing of gonadectomy (GDX) or sham surgery, cognitive assessment by Morris Water Maze (MWM), in vivo MRI, and pathology (PATH) at young, midlife, and old ages. **b** In vivo MRI was collected from young, midlife, and old female and male mice. Substructure volumes visualized on the mean template (dorsal hippocampus = red, ventral hippocampus = green, cortex = blue, striatum = purple). Sham-treated female (red) and male (blue) volumes are expressed as a percentage of intercranial volume (ICV) for (**c**) frontal cortex, (**d**) striatum, and (**e**) dorsal hippocampus over the lifespan. Males showed gradual atrophy in frontal cortex and striatum from young to midlife to old age. In contrast, females maintained volumes through midlife, followed by atrophy from midlife to old age. Dorsal hippocampus showed atrophy in females from midlife to old age, while males did not have atrophy. Two-way ANOVA indicated a significant interaction between sex and age in dorsal hippocampus ($p = 0.0059$). $n = 12$ for all groups (**c**–**e**). $p$ values were calculated by two-sided Welch's $t$-test. Female sham-treated (solid) and GDX (diagonal lines) substructure volumes, assessed by MRI, expressed as a percentage of intercranial volume (ICV) are shown for (**f**) whole hippocampus, (**g**) dorsal hippocampus, and (**h**) ventral hippocampus. GDX females showed smaller dorsal hippocampus than sham females at midlife ($p = 0.014$). Female midlife sham $n = 6$ and midlife GDX $n = 8$, old sham $n = 5$ and old GDX $n = 8$. $p$ values were calculated by two-sided Welch's $t$-test. All box plots with center lines showing the medians, boxes indicating the interquartile range, and whiskers indicating a maximum of 1.5 times the interquartile range beyond the box.

performance). Correlations using hippocampal substructures demonstrated that this effect in whole hippocampus was driven by a positive correlation between time spent in TQ and the volume of dorsal hippocampus (Fig. 2e), not ventral hippocampus (Fig. 2f). This suggested that dorsal hippocampal atrophy as measured by in vivo MRI in mice is a sensitive biomarker for hippocampal-dependent cognitive impairment in aging. Also, these functional correlations underscored the importance of a region-specific approach in evaluating brain atrophy during aging.

### Dorsal hippocampal neuropathology

Given the results in dorsal hippocampal-dependent behavioral test performance, we next assessed the effect of GDX on neuropathology within dorsal hippocampus. Reactive astrogliosis was assessed by colocalization of LCN2 and GFAP. An increase in LCN2+GFAP+ astrocytes was observed in GDX female mice compared to gonadally intact (sham) females at midlife (Fig. 3a, b). In contrast, there was no effect of GDX in young mice (Fig. 3b). Next, microglia activation was assessed by colocalization of IBA1 and MHCII. There was an increase in IBA1+MHCII+ microglia in GDX females compared to gonadally intact at midlife (Fig. 3c, d), but not at young age (Fig. 3d). Disease-associated microglia (DAM) were assessed by colocalization of CLEC7A and P2RY12, as described[50]. An increase in DAM was observed in GDX female mice compared to gonadally intact at midlife (Fig. 3e, f), but not at young age (Fig. 3f). Lastly, synaptic loss was assessed using co-localization of expression of the pre-synaptic marker SYN1 and the post-synaptic marker PSD95. Synaptic loss was observed in GDX female mice compared to gonadally intact at midlife (Fig. 3g, h), but not at young age (Fig. 3h). Together, neuropathology of dorsal hippocampus revealed a sex hormone by age interaction, whereby loss of ovarian sex hormones alone or aging to midlife alone was not deleterious. Instead, both sex hormone loss and aging were needed to induce glial activation and synaptic loss in dorsal hippocampus in female mice.

To determine the relationship between spatial reference memory and hippocampal pathology, we correlated time spent in the TQ on MWM testing with neuropathology outcomes. There was a negative correlation between time spent in the TQ with LCN2+GFAP+ astrocytes ($r = -0.46203$, $p = 0.017$) as well as with MHCII+IBA1+ microglia

hormones in female mice induced cognitive impairment at midlife, but not young age.

### Correlations between dorsal hippocampal volumes by in vivo MRI and performance on Morris Water Maze testing

To determine the relationship between hippocampal atrophy and spatial reference memory, we correlated hippocampal substructure volumes with time spent in the target quadrant (TQ) on the MWM. We observed that there was a positive correlation between whole hippocampal volume and time spent in TQ ($r = 0.22$, $p = 0.049$). Specifically, smaller volumes (worse atrophy) in the hippocampus were associated with less time in the TQ (worse spatial reference memory

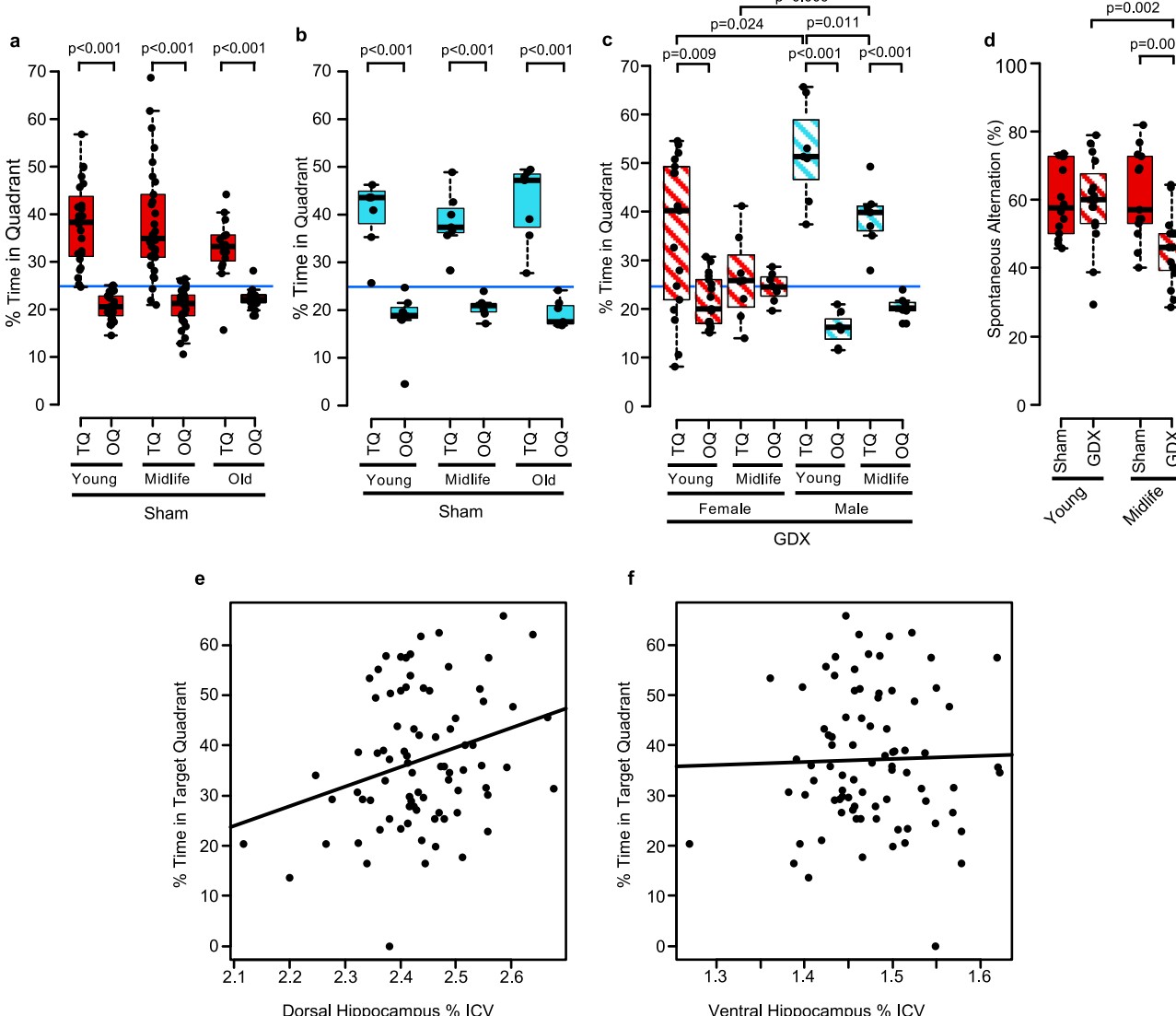

**Fig. 2 | Ovariectomy of female mice worsens cognition at midlife, but not young age. a, b** Cognitive assessment by the Morris water maze. Three different ages (Young, Midlife, and Old) of sham-treated C57BL/6 wild type (**a**) females or (**b**) males were tested using Morris Water Maze (MWM) behavioral testing. A probe trial was conducted 2 h after the 5-day platform hidden training (without escape plat-form). % of time in the target quadrant of the maze (TQ) indicated memory of the platform location in preference to the average of other 3 quadrants (OQ).
**a** Significant preference for the TQ compared to the OQ for all ages was observed in sham treated females ($p < 0.0001$, TQ vs. OQ in all ages), indicating intact reference memory in each group with no between group differences. Blue line indicates the null hypothesis (25% in TQ). Female young $n = 24$, midlife $n = 30$, and old $n = 16$.
**b** Significant preference for the TQ compared to the OQ for all ages was observed in sham treated males indicating intact reference memory in each group. Male young $n = 7$, midlife $n = 14$, and old $n = 7$. **c** Percent (%) time in target quadrant (TQ) and the average of other quadrants (OQ) among females (red) and males (blue), at young and midlife, all GDX (diagonal lines). Intact cognition (significant preference for the

TQ compared to OQ) was observed in GDX females young and in all GDX males (young and midlife), while GDX females at midlife had impairment. Between groups differences showed a significant decrease of % time in TQ in GDX females at midlife compared to GDX males at midlife ($p = 0.0093$). There was also a significant decrease of % time in TQ in GDX females at midlife compared to sham females at midlife (**c** vs. **a**, $p = 0.0085$). Female GDX young $n = 17$ and midlife $n = 8$, Male GDX young $n = 7$ and midlife $n = 7$. **d** Significant working memory impairment, assessed by Y maze, was observed in midlife GDX females ($p = 0.0011$, vs. midlife sham; $p = 0.0024$, vs. young GDX). Female young sham $n = 14$ and GDX $n = 16$, midlife sham $n = 13$ and GDX $n = 16$. Percent (%) time in TQ correlated with (**e**) dorsal hippocampus volume ($r = 0.28$; $p = 0.011$), but not with (**f**) ventral hippocampus volume ($r = 0.027$; $p = 0.809$). $p$ values were calculated by two-sided Mann–Whitney $U$ test, except for two-sided Pearson correlation analyses (**e, f**). All box plots with center lines showing the medians, boxes indicating the interquartile range, and whiskers indicating a maximum of 1.5 times the interquartile range beyond the box.

($r = -0.4151$, $p = 0.035$) (Supplementary Fig. 3). Specifically, worse spatial reference memory performance was associated with increased activation of astrocytes and microglia. Similar correlations were also observed for working memory using Y maze, with a negative correlation between spontaneous alternations with LCN2$^+$GFAP$^+$ astrocytes ($r = -0.51211$, $p = 0.006$) as well as with MHCII$^+$IBA1$^+$ microglia ($r = -0.47227$, $p = 0.017$). Thus, spatial reference memory and working memory performance were each associated with glial activation in dorsal hippocampus.

## Selective deletion of ERβ in astrocytes and neurons

Next, we investigated a cell-specific and hormone receptor-specific mechanism through which endogenous estrogens could be mediating dorsal hippocampal neuroprotection. Hippocampal astrocytes play a role in memory formation and synaptic transmission[30,45,51,52]. ERβ is expressed in both astrocytes and neurons[53–55]. In addition, ERβ has potential as a therapeutic target since its activation does not induce ERα mediated adverse effects on breast. Thus, we next determined whether selective deletion of ERβ in either astrocytes or neurons in

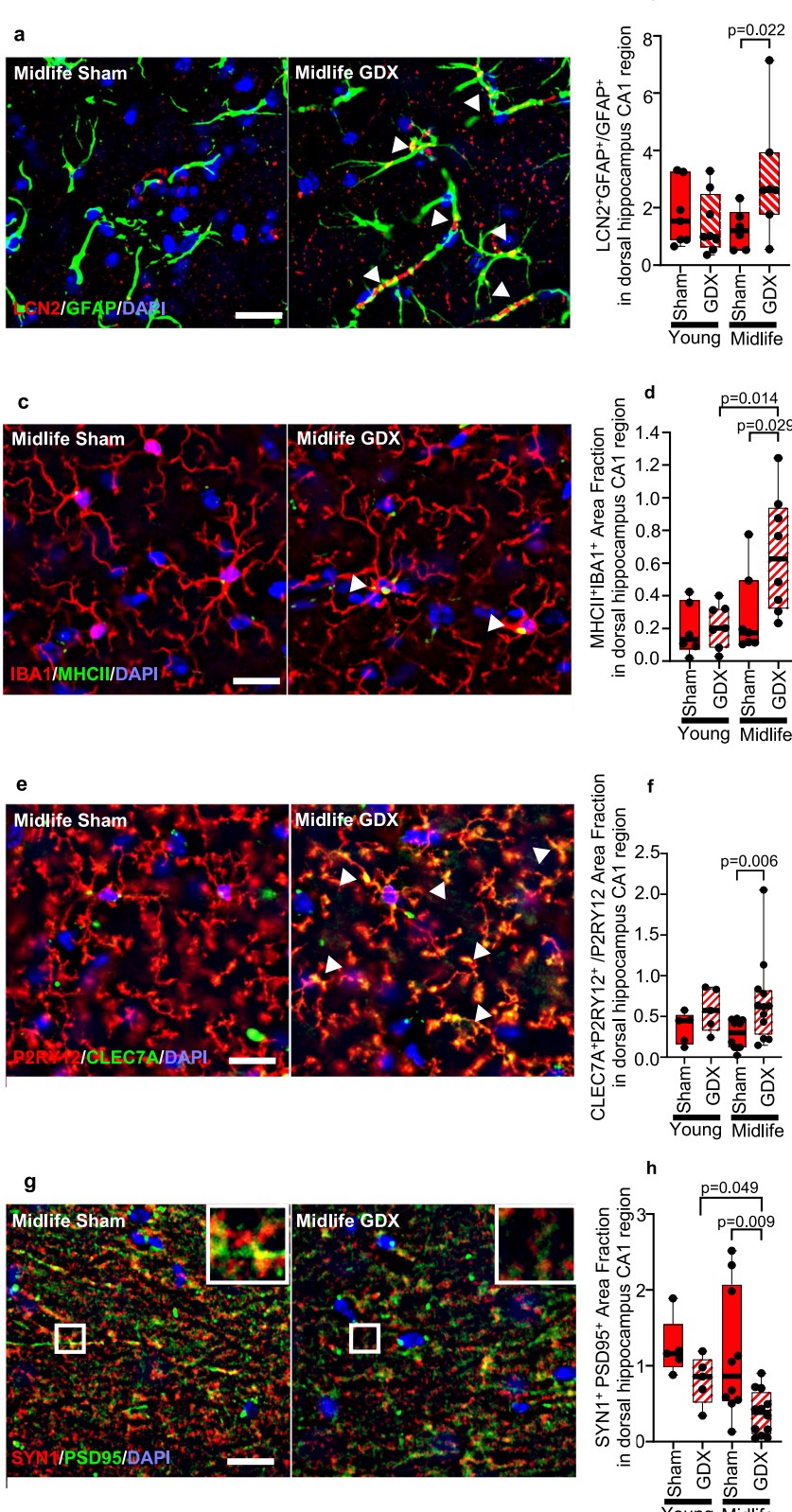

gonadally intact female mice could recapitulate the effect of loss of endogenous estrogen during gonadectomy on dorsal hippocampal outcomes. To this end, we created estrogen receptor β (ERβ) conditional knock-outs in astrocytes (astrocyte ERβ cKO) or neurons (neuron ERβ cKO). To generate the astrocyte ERβ cKO and the neuron ERβ cKO, we chose the *mGFAP-Cre 77.6* line[56] and the rat *neuronal specific*

*enolase (NSE)-Cre* line[57,58], respectively. Each was crossed with mice containing homozygous floxed *Esr2* alleles. The breeding scheme drove Cre recombinase allele inheritance from females (mother), not from males (father), to avoid ectopic expression of Cre recombinase in the male germline (see Methods). Validation of ERβ deletion in the neuron ERβ cKO was previously published[57]. To investigate the

**Fig. 3 | Gonadectomy induces glial activation and synaptic loss in dorsal hippocampus in female mice at midlife, but not young age. a** Representative ×40 images of LCN2 (red), GFAP (green), merged images for colocalization (yellow, white arrowheads) showing astrocyte reactivity; **c** MHCII (green), IBA1(red), merged images for colocalization (yellow, white arrowheads) showing activated microglia; **e** CLEC7A (green), P2RY12 (red), merged image for colocalization (yellow, white arrowheads) showing disease-associated microglia (DAM); **g** SYN1(red), PSD95 (green), merged images for colocalization (yellow, white arrowheads) showing pre- and post-synaptic staining. Inset: Magnification at ×100. Nuclei were counterstained with DAPI (blue). Bar = 20 um. Quantitative analysis of (**b**) LCN2$^+$GFAP$^+$ area fraction, (**d**) MHCII$^+$IBA1$^+$ area fraction, (**f**) CLEC7A$^+$P2RY12$^+$ area fraction, and (**h**) SYN1$^+$PSD95$^+$ area fraction in dorsal hippocampal CA1 region from GDX and sham females at young and midlife ages. Midlife GDX females showed a significant increase in reactive astrocytes ($p = 0.0221$), activated microglia ($p = 0.0289$), DAM ($p = 0.0028$), and synaptic loss ($p = 0.009$) each as compared to midlife sham females. Female young sham $n = 7, 6, 5, 5$, and GDX $n = 8, 7, 5, 5$, midlife sham $n = 6, 7, 10, 10$ and GDX $n = 7, 8, 12, 12$ (**b, d, f, h**, respectively). $p$ values were calculated by two-sided Mann–Whitney $U$ test. All box plots with center lines showing the medians, boxes indicating the interquartile range, and whiskers indicating from the minimum and to the maximum values.

specificity of Cre expression in the *mGFAP-Cre 77.6* line, we crossed the *mGFAP-Cre 77.6* line to RiboTag mice (Fig. 4a) and assessed HA expression in hippocampus using immunofluorescence staining. Double immunolabeling showed colocalization of HA with astrocyte markers GFAP and ALDH1L1 (Fig. 4b, c). Lack of colocalization was observed with other cell-specific markers, namely NEUN for neurons, P2RY12 and IBA1 for microglia, and CC1 for oligodendrocytes (Fig. 4b). Following confirmation of astrocyte-specific Cre expression in hippocampus of the *mGFAP-Cre 77.6* line, we crossed the *mGFAP-Cre 77.6* line with mice containing homozygous floxed *Esr2* alleles (Fig. 4d). Efficacy of selective deletion of ERβ in hippocampal astrocytes was shown by immunofluorescence staining (Fig. 4e). ERβ$^+$ GFAP$^+$ colocalized staining was significantly reduced in the astrocyte ERβ cKO as compared to that in wild type (WT) littermates and in neuron ERβ cKO mice (Fig. 4f). Conversely, ERβ expression remained intact in neurons of the astrocyte-ERβ cKO (see Supplementary Fig. 4).

**Neuroprotection in female mice at midlife is mediated by ERβ in astrocytes**

At midlife, gonadally intact mice with selective deletion of ERβ in either astrocytes or neurons, as well as WT littermates, underwent cognitive behavioral testing and in vivo MRI to determine whether a protective effect of endogenous estrogens on hippocampal atrophy in females at midlife was mediated through ERβ in astrocytes or neurons. In the MWM task, WT female mice showed significant preference for the TQ compared to other quadrants (OQ). In contrast, astrocyte ERβ cKO females did not show significant preference for the TQ (Fig. 5a). Neuron ERβ cKO female mice demonstrated no cognitive deficit thereby underscoring the importance of cell specificity in ERβ gene deletion.

Regarding regional brain atrophy by in vivo MRI, astrocyte ERβ cKO mice showed atrophy in the hippocampus compared to WT littermates, while neuron ERβ cKO mice did not have atrophy (Fig. 5b). This atrophy in whole hippocampus in astrocyte ERβ cKO mice was primarily driven by the effect in the dorsal hippocampus (Fig. 5c), not the ventral hippocampus (Fig. 5d). This corroborated previous observations in GDX wild type mice where whole hippocampal atrophy was driven by atrophy in dorsal, not ventral, hippocampus (Fig. 1f–h).

The effect of selective deletion of ERβ in either astrocytes or neurons on dorsal hippocampal neuropathology in midlife female mice was then examined. Reactive astrocytes assessed by LCN2 and GFAP (Fig. 5e, f) and activated microglia assessed by IBA1 and MHCII (Fig. 5g, h) demonstrated increased expression in dorsal hippocampus in the astrocyte ERβ cKO compared to WT littermates. In addition, an increase in disease-associated microglia (DAM) was measured by CLEC7A and P2RY12 was observed in the astrocyte ERβ cKO compared to WT littermates (Fig. 5i, j). Lastly, synaptic loss as measured by co-localization of SYN1 and PSD95 expression was found in the astrocyte ERβ cKO compared to WT littermate (Fig. 5k, l). In contrast to these findings in the astrocyte ERβ cKO, the neuron ERβ cKO mice did not show differences in neuropathology as compared to WT littermates, again underscoring the importance of cell specificity in ERβ gene deletion.

In contrast to above observations in midlife female mice, we then asked whether cognitive impairment or hippocampal atrophy was present in astrocyte ERβ cKO mice that were young, and we found that

this was not the case. Young (age 3–4 month) female astrocyte ERβ cKO, as compared to WT littermates, had no deficits in spatial reference memory (Supplementary Fig. 5a). There was also no hippocampal atrophy in young astrocyte ERβ cKO as compared to young WT littermates (Supplementary Fig. 5b–d).

**Hippocampal astrocyte transcriptomes in midlife female mice that have selective deletion of ERβ in astrocytes**

Since two complementary loss-of-function studies showed causality whereby decreased ERβ ligation induced neurodegenerative changes in dorsal hippocampus in female mice at midlife, we next determined the effect on the transcriptome of hippocampal astrocytes. To this end, we applied RiboTag technology and RNA sequencing analyses as previously implemented by our lab[32,59,60], here using the unique astrocyte-RiboTag mice characterized in Fig. 4. Further, we crossed male astrocyte RiboTag mice with female astrocyte ERβ cKO mice to create astrocyte-RiboTag mice with selective deletion of ERβ in astrocytes. This revealed changes in gene expression in hippocampal astrocytes caused by targeted deletion of ERβ in astrocytes. Sham treated astrocyte ERβ cKO midlife female mice had decreased expression of estrogen response element (ERE) harboring genes in hippocampal astrocytes compared to sham treated WT midlife females (Fig. 6a). Providing context for this finding in targeted genetic manipulation, this decrease in ERE harboring genes was also found in hippocampal astrocytes from WT (GDX) compared to WT (sham). Indeed, five of the seven genes most decreased in expression were genes known to be estrogen responsive[61,62].

Genome wide effects of selective deletion of ERβ in hippocampal astrocytes were determined using pathway analyses. The top two most differentially expressed pathways in hippocampal astrocytes derived from astrocyte-ERβ cKO compared to WT mice were the Gluconeogenesis I and Glycolysis I pathways (Fig. 6b). The top most differentially expressed gene in these two pathways was *enolase 1* (*Eno1*), an enzyme responsible for the reversible conversion of phosphoenolpyruvate to phosphoglycerate during gluconeogenesis and glycolysis. The direction of change was an increase in the astrocyte-ERβ cKO (Fig. 6c). Menopause and ovariectomy have each been shown to cause poor glucose utilization and increased gluconeogenesis in liver hepatocytes[63–66]. While gluconeogenesis is known to occur in brain astrocytes[67,68], an effect of ERβ in astrocytes on the gluconeogenesis pathway in astrocytes has not been previously reported.

**Upregulation of human *Enolase 1* gene expression in hippocampus of women during aging**

For translational relevance, we next examined *ENO1* expression in human hippocampal data in the GEO database. We found a direct correlation between higher *ENO1* expression and older ages in women ($r = 0.526$, $p = 0.014$) (Fig. 7a). Postmenopausal age women (>60 years of age) had higher *ENO1* levels in hippocampus compared to younger women (<50 years of age) (Fig. 7b). Together, our finding of higher *ENO1* expression in hippocampus of menopausal age women (>60 years) (Fig. 7b) is consistent with higher *Eno1* expression in hippocampal astrocytes of midlife female mice with specific deletion of ERβ in astrocytes (Fig. 6c).

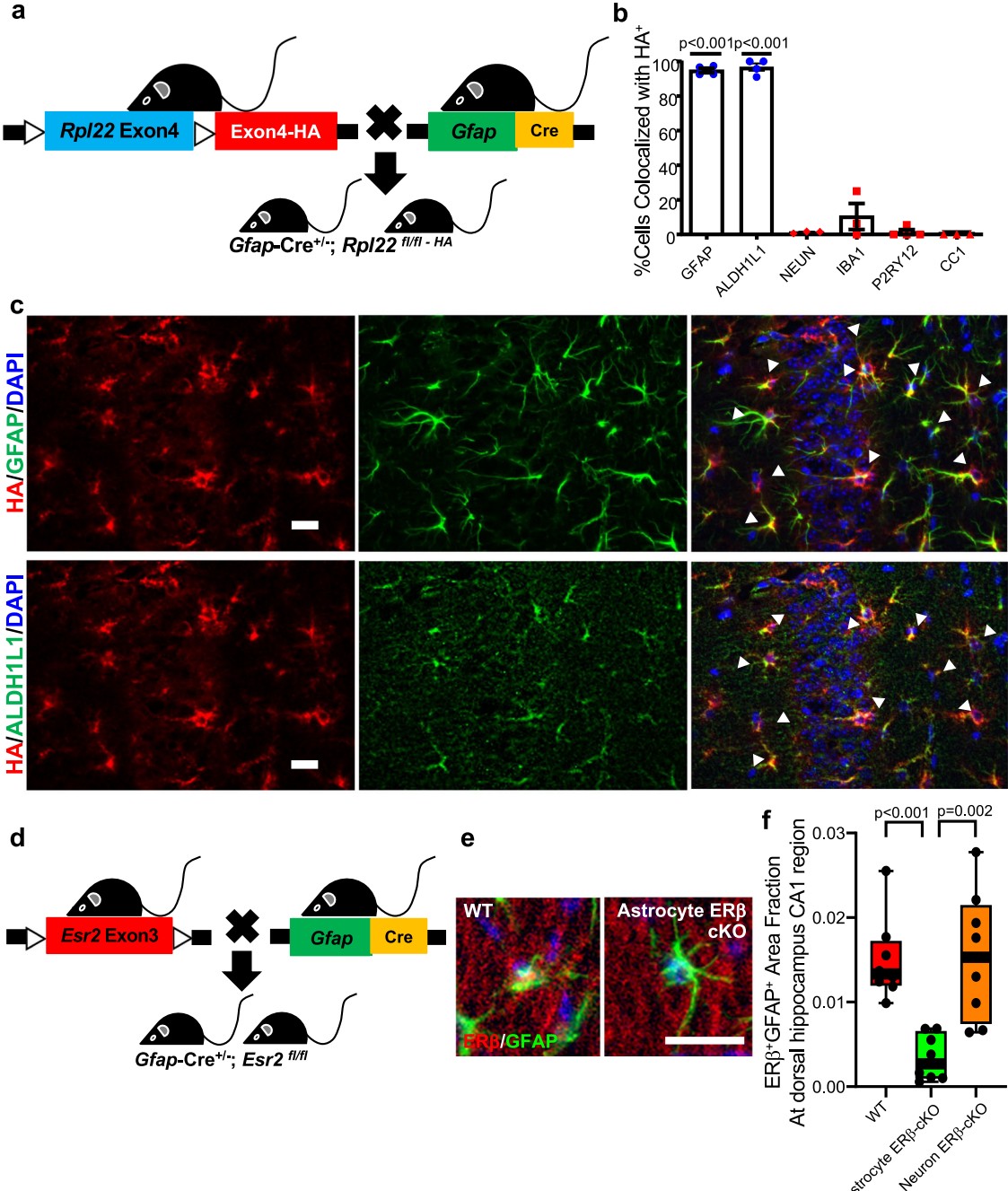

**Fig. 4 | Validation of Cre expression specificity in *mGFAP-Cre 77.6* mice and ERβ selective deletion in the astrocyte ERβ cKO. a** Breeding scheme to generate mGFAP-Cre; RiboTag mice. **b** Quantification of HA-colocalization with GFAP+ and ALDH1L1+ for astrocytes, NEUN+ cells for neurons, IBA1+ and P2RY12+ for microglia, and CC1+ cells for oligodendrocytes. $N = 4$ for GFAP, ALDH1L1, and P2RY12, and $n = 3$ for IBA1, NEUN, and CC1. Error bars represent SEM. *p* values were calculated by two-sided one sample *t*-test. **c** Representative ×40 images of HA (red), GFAP (green, top), and ALDH1L1 (green, bottom) with merged images for colocalization (yellow, arrowheads) in dorsal hippocampal CA1 region from *mGFAP-Cre(77.6); RiboTag* mice. Bar = 20 um. Experiments were repeated independently at least twice. **d** Breeding scheme to generate *mGFAP-Cre(77.6); ERβ^fl/fl* mice. **e** Representative ×40 image of ERβ (red) and GFAP (green) with colocalization (yellow) in dorsal hippocampus CA1 region from WT littermates (WT, left), *mGFAP-Cre(77.6); ERβ^fl/fl* (astrocyte ERβ cKO, right) mice. Bar = 10 um. **f** Quantitative analysis of area fraction of colocalized ERβ+ GFAP+ in dorsal hippocampal CA1 region from WT littermates (WT, red), astrocyte ERβ cKO (green), and *rNSE-Cre; ERβ^fl/fl* (neuron ERβ cKO, orange) mice. A significant decrease of ERβ+ GFAP+ colocalized area was observed in astrocyte ERβ cKO, compared to WT littermates ($p = 0.0002$) and compared to neuron ERβ cKO ($p = 0.0019$). $n = 8$ per group. *p* values were calculated by two-sided Mann–Whitney *U* test. Box plots with center lines showing the medians, boxes indicating the interquartile range, and whiskers indicating from the minimum and to the maximum values.

## ERβ-ligand treatment reverses dorsal hippocampal neuropathology in midlife female mice

Finally, since loss-of-function studies showed that not only GDX of female mice, but also selective deletion of ERβ in astrocytes in gonadally intact females, caused hippocampal-dependent cognitive impairment and dorsal hippocampal pathology at midlife, we next performed gain-of-function studies. WT female mice were GDX or sham treated at 11 months of age, and 1 week later treated with either ERβ ligand or vehicle for a duration of 1 month, before undergoing MWM behavioral testing and dorsal hippocampal pathology analyses

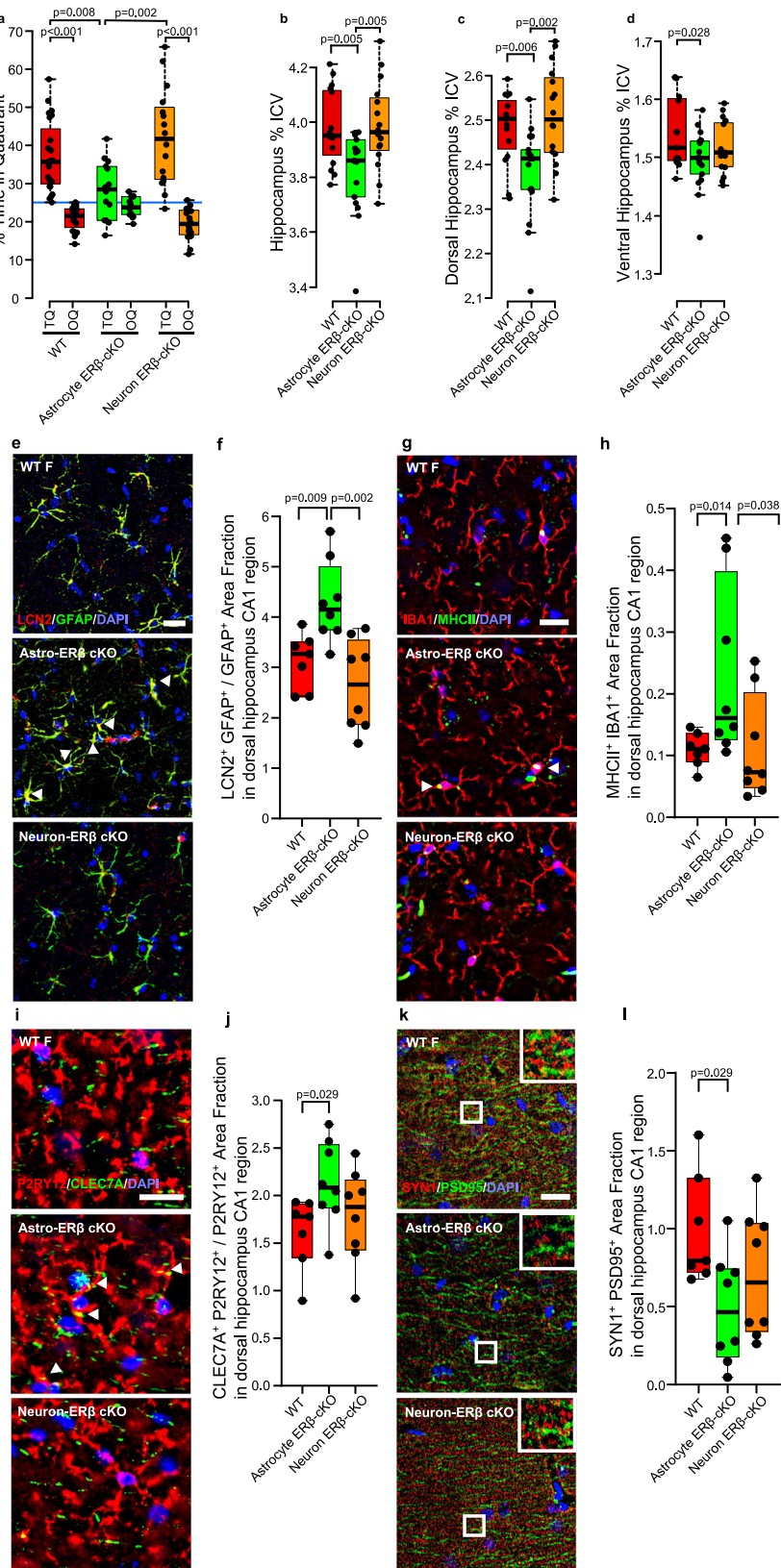

(see schematic, Fig. 8a). As expected, midlife sham treated female mice showed significant preference for TQ over OQ. In GDX mice that were vehicle treated, there was no longer significant preference for the TQ. In contrast, GDX female mice at midlife that were ERβ ligand treated showed significant preference for the TQ over OQ (Fig. 8b). Regarding dorsal hippocampal neuropathology, midlife GDX female mice

demonstrated significantly increased LCN2⁺GFAP⁺ reactive astrocytes, MHCII⁺IBA1⁺ activated microglia, and CLEC7A⁺P2RY12⁺ disease-associated microglia, with reduced SYN1⁺PSD95⁺ synaptic staining, each as compared to midlife sham females (Fig. 8c–f, diagonal lines). Importantly, this astrocyte and microglial activation and synaptic loss were reversed in midlife GDX females treated with ERβ ligand

**Fig. 5 | Selective deletion of ERβ in astrocytes induces cognitive deficits, dorsal hippocampal atrophy and neuropathology therein at midlife in female mice.** **a** Gonadally intact wild type (WT, red) females along with conditional knock-outs of ERβ in astrocytes (astrocyte ERβ cKO, green) or in neurons (neuron ERβ cKO, orange) were assessed at midlife for spatial reference memory by MWM. WT and neuron ERβ cKO showed preference for the TQ over the OQ ($p < 0.0001$), while astrocyte ERβ cKO did not. Blue line indicates the null hypothesis (25% in TQ). Between groups differences showed a significant decrease in % time in TQ in gonadally intact females with ERβ deleted in astrocytes ($p = 0.0077$ vs. WT; $p = 0.0021$ vs. neuron ERβ cKO). WT $n = 23$, astrocyte ERβ cKO $n = 14$, and neuron ERβ cKO $n = 16$. Substructure volumes, assessed by MRI, taken as a percentage of intercranial volume (ICV) are shown for (**b**) hippocampus, (**c**) dorsal hippocampus, and (**d**) ventral hippocampus. Astrocyte ERβ cKO females showed worse atrophy compared to WT and compared to neuron ERβ cKO at midlife in whole hippocampus ($p = 0.0048$, vs. WT; $p = 0.004973$, vs. neuron ERβ cKO) and dorsal hippocampus ($p = 0.0055$ vs. WT; $p = 0.0016$ vs. neuron ERβ cKO). WT $n = 15$, astrocyte ERβ cKO $n = 17$, and neuron ERβ cKO $n = 18$. **e**–**l** Neuropathology. Representative ×40 images of (**e**) LCN2 (red), GFAP (green), merged images for colocalization (yellow, white arrowheads) showing astrocyte reactivity, (**g**) MHCII (green),

IBA1(red), merged images for colocalization (yellow, white arrowheads) showing activated microglia, (**i**) CLEC7A (green), P2RY12 (red), merged image for colocalization (yellow, white arrowheads) showing disease-associated microglia (DAM), (**k**) SYN1 (red), PSD95 (green), merged images for colocalization (yellow, white arrowheads) showing pre- and post-synaptic staining. Inset: Magnification at ×100. Nuclei were counterstained with DAPI (blue). Bar = 20 um. Quantitative analysis of (**f**) LCN2⁺GFAP⁺ area fraction for reactive astrocytes, (**h**) MHCII⁺IBA1⁺ area fraction for activated microglia, (**j**) CLEC7A⁺P2RY12⁺ area fraction for disease associated microglia (DAM), and (**l**) SYN1⁺PSD95⁺ area fraction for synapses in dorsal hippocampal CA1 region among WT (red), astrocyte ERβ cKO (green), and neuron ERβ cKO (orange) females at midlife. Astrocyte ERβ cKO showed a significant increase of reactive astrocytes ($p = 0.0093$, vs. WT; $p = 0.0019$, vs. neuron ERβ cKO), activated microglia ($p = 0.014$, vs. WT; $p = 0.0379$, vs. neuron ERβ cKO), and DAM ($p = 0.0289$, vs. WT), as well as synaptic loss ($p = 0.0289$, vs. WT). WT $n = 7$, astrocyte ERβ cKO $n = 8$, and neuron ERβ cKO $n = 8$. $p$ values were calculated by either two-sided Mann–Whitney $U$ test (**a**, **f**, **h**, **j**, **l**) or two-sided Welch's $t$-test (**b**–**d**). All box plots with center lines showing the medians, boxes indicating the interquartile range, and whiskers indicating either a maximum of 1.5 times the interquartile range beyond the box (**a**–**d**) or from the minimum and to the maximum values (**f**, **h**, **j**, **l**).

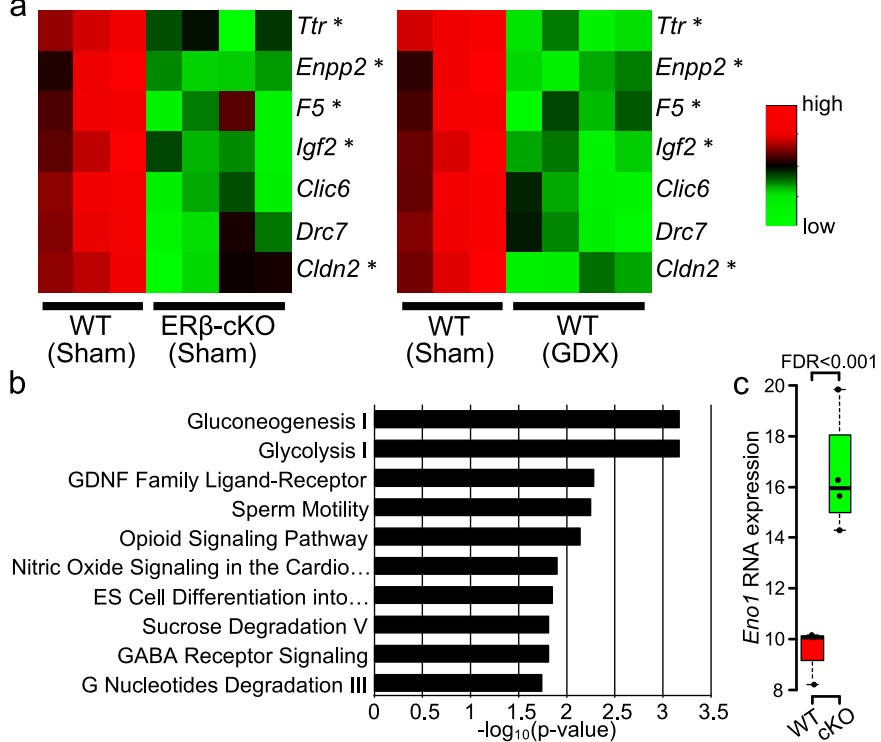

**Fig. 6 | RNA sequencing and pathway analysis of hippocampal astrocytes from female astrocyte-ERβ cKO mice at midlife. a** Heatmaps of top seven down-regulated genes (FDR < 0.1, log2 fold change is >0.5) in hippocampal astrocytes from astrocyte ERβ cKO mice and corresponding genes from GDX mice, each compared to that observed in WT sham mice. Asterisks indicate genes previously reported as estrogen responsive[61,62]. **b** Top two most differentially expressed pathways in dorsal hippocampal astrocytes of ERβ cKO versus WT mice were the

Gluconeogenesis I and Glycolysis I pathways. Right-tailed Fisher's exact test. **c** The top differentially expressed gene in the Gluconeogenesis I and Glycolysis I pathways was *Enolase 1* (*Eno1*) in hippocampal astrocytes of ERβ cKO versus WT mice. (WT, $n = 3$; cKO, $n = 4$, FDR = 0.00036), two-sided Mann–Whitney $U$ test. For the box-and-whisker plots, the box indicates the median and 25–75th percentile range, and the whiskers extend to a maximum of 1.5 times the interquartile range beyond the box.

(Fig. 8c–f, crossed lines). Together, this revealed reversibility of dorsal hippocampal pathology using treatment targeting ERβ in otherwise healthy female mice at midlife.

## Discussion

Here, we found a sex difference in the trajectory of regional brain atrophy by in vivo MRI in female versus male mice from young, to midlife, to old ages. Females had relative protection from substructure atrophy at midlife, followed by an abrupt decline, while males had gradual atrophy across the lifespan. Gonadectomy also revealed a sex

difference. Gonadal hormone loss induced hippocampal-dependent memory impairment in females, but not males, at midlife. That said, loss of endogenous ovarian hormones in gonadectomized female mice did not cause cognitive impairment at young ages, revealing a sex hormone by age interaction. Gonadectomy in female mice also induced an increase in glial activation and synaptic loss in dorsal hippocampus at midlife, but not at young ages, again revealing a sex hormone by age interaction, this time in neuropathology.

Beneficial effects of estrogens on cognitive function and hippocampal synaptic pathology have been described for decades, but

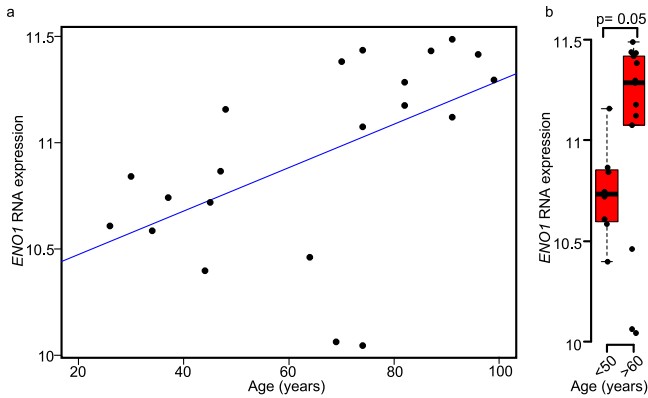

**Fig. 7 | Upregulation of human *Enolase 1* gene in hippocampus in women during aging. a** Analysis of human hippocampus microarray data of in women (GSE11882, ages: 26–99 years old, $n = 21$). Higher gene expression levels of *Enolase 1* (*ENO1*) correlated with older ages ($r = 0.526$; $p = 0.014$, blue line). **b** Human *ENO1* gene expression was increased in females at menopausal ages (>60 years, $n = 13$) compared to younger ages (<50 years, $n = 8$), two-sided Mann–Whitney *U* test. For the box-and-whisker plots, the box indicates the median and 25–75th percentile range, and the whiskers extend to a maximum of 1.5 times the interquartile range beyond the box.

which hippocampal cell type or estrogen receptor subtype is essential for in vivo neuroprotection in otherwise healthy female mice at midlife has remained unclear. Here, we used gonadally intact female mice with selective deletion of ERβ in either astrocytes or neurons to determine if ERβ in either of these cells is required for neuroprotection in females at midlife. Deletion of ERβ in astrocytes, but not neurons, induced hippocampal-dependent memory impairment, dorsal hippocampal atrophy by in vivo MRI, and dorsal hippocampal glial activation and synaptic loss in female mice at midlife. This region-specific, cell-specific, and hormone receptor-specific approach identified ERβ in astrocytes for precision targeting to prevent cognitive deficits in female mice at midlife, with potential translational relevance for understanding deficits in cognitive domains aligning with posterior hippocampal function in early menopausal women at midlife.

The menopausal transition aligns with deleterious effects on cognitive performance and dysregulation of glucose metabolism in brains of women at midlife, and estrogen therapy aiming to restore glucose utilization and brain energetics with less dependence on gluconeogenesis has been proposed for these women[69–73]. However, studies in preclinical models are needed to investigate brain cell-specific and estrogen receptor-specific mechanisms. Menopause and ovariectomy induce poor glucose utilization and increase gluconeogenesis in liver hepatocytes[63–66], and gluconeogenesis is known to occur in brain astrocytes[67,68]. Yet the effect of menopause and ovariectomy on gluconeogenesis and glycolysis in astrocytes in females at midlife remains understudied. Our genome wide pathway analyses showed these two metabolic pathways as the most differentially expressed in hippocampal astrocytes of midlife female mice with selective deletion of ERβ in astrocytes. The increase in *Eno1* was the top most differentially expressed gene in these pathways in mice. *ENO1* was also increased in the hippocampus of menopausal women. Further studies are now warranted focusing on ERβ mediated regulation of glucose metabolism in astrocytes of midlife female mice and humans.

The *Apolipoprotein E* (*APOE*) gene in humans is a major risk factor for cognitive decline in aging, with the *APOE4* allele conferring risk compared to *APOE3*[69]. Astrocytes are the main source of APOE in brain, providing metabolic substrates, such as lactate, to neurons as a means of glial energy distribution. This astrocyte-neuron metabolic coupling is important for neuronal function[67,74,75]. Previous in vitro studies found that *APOE4*, compared to *APOE3*, astrocytes had impaired glucose uptake with increased gluconeogenesis[67]. Also, the effects of ERβ ligand

treatments in preclinical AD models have suggested more benefit in the presence of *APOE3* compared to *APOE4* alleles[76,77]. In women and female mice, estrogen treatment has shown more benefit on cognitive function in the context of *APOE3* compared to *APOE4* alleles[78,79]. Together, this prompts speculation of a two hit hypothesis. A decrease in protective ERβ ligation and a deleterious effect of *APOE4* inheritance on glucose metabolism pathways in astrocytes may combine to induce hippocampal dependent cognitive decline in menopausal women carrying the *APOE4* allele. This potential sex hormone by *ApoE* gene interaction in hippocampal astrocytes warrants further investigation.

While selective deletion of ERβ in astrocytes, but not neurons, was identified as a cell-specific and receptor-specific therapeutic target for cognitive decline in female mice at midlife, this is not mutually exclusive of an additional target cell. For example, ERβ ligand treatment induced neuroprotection in the MS preclinical model through an effect in CD11c$^+$ cells of the microglia/macrophage lineage, and this was not mutually exclusive of an additional role of ERβ in Olig1$^+$ cells of oligodendrocyte lineage[59,80]. Selective interruption of either mechanism abrogated ERβ ligand treatment mediated neuroprotection in spinal cord. Here, an effect of ERβ in microglia in dorsal hippocampus during menopause remains possible. Indeed, astrocytes play a role in synaptic loss and function through direct effects on maintaining synaptic health and through indirect effects on microglia and subsequent synaptic engulfment and plasticity[30,45,81–86]. Like astrocytes, microglia can have beneficial or deleterious functions depending on brain region and disease process[51]. Thus, selective deletion of ERβ in microglia in female mice at midlife is needed to determine the effect on hippocampal-dependent cognitive impairment, dorsal hippocampal atrophy by in vivo MRI, and dorsal hippocampal neuropathology.

Looking forward to potential translation of these preclinical findings in mice, one can use a new lens focused on specificity when reviewing past clinical research. Cognitive problems during natural menopause at midlife have been known for decades. However, repurposing standard hormone replacement therapy (HRT) using conjugated equine estrogens (Premarin) or estradiol at doses and durations to treat hot flashes, genitourinary, and other menopausal symptoms has not demonstrated neuroprotection[7,87]. The "timing hypothesis" states that estrogen treatment may be generally more effective when started at early menopausal ages compared to after age 65 years. Still, cognition was neither improved nor worsened by treatment with low dose estradiol in women within 6 to 10 years of menopause[7]. Thus, while timing is important, attention to only this appears insufficient to achieve neuroprotection[87]. That said, previous research using treatment with estradiol, but not Premarin, in early menopausal women showed some positive signals if one considers brain regional effects, namely on posterior hippocampus and prefrontal cortex[5,6,13,88,89]. Conversely, being female and menopausal conferred risk for having regions of brain atrophy and lower glucose utilization by (18)F-fluorodeoxyglucose (FDG) PET in subjects ages 40 to 65 years[90]. This is intriguing given findings here of gene expression changes involving glucose metabolism in hippocampal astrocytes of midlife female mice with ERβ deleted in astrocytes (Fig. 6) as well as in hippocampus of menopausal women (Fig. 7).

The pregnancy hormone estriol binds preferentially to ERβ over ERα[91,92]. Thus, preclinical findings here using cell-specific deletion of ERβ in female mice (Fig. 5), transcriptomics of astrocytes when ERβ is selectively deleted (Fig. 6), and ERβ ligand treatment in female mice (Fig. 8) warrant discussion with respect to a Phase 2 trial using treatment with estriol in women with MS, ages 18–50 years[93]. Estriol treated MS women, compared to placebo treated, showed an improvement in performance on cognitive tests of processing speed, and better cognitive performance correlated with higher estriol blood levels[93]. Estriol treated MS women had less atrophy of subregions in cerebral cortex and also showed beneficial effects on other biomarkers of neurodegeneration[93–95].

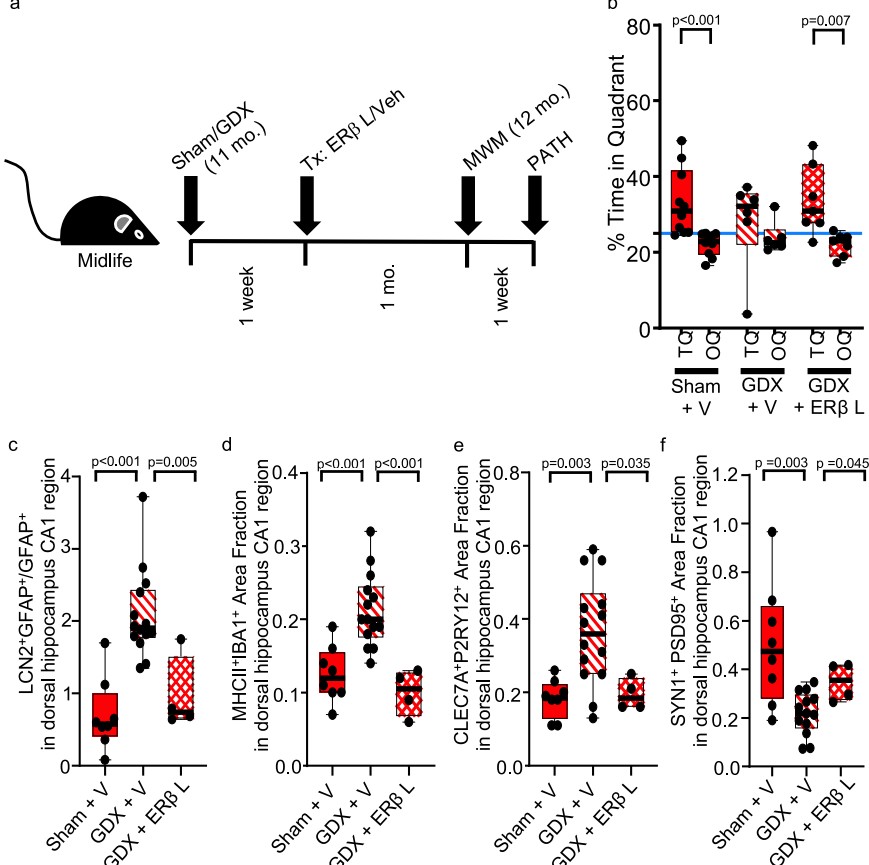

**Fig. 8 | ERβ ligand treatment reverses dorsal hippocampal pathology in female mice at midlife. a** Schematic showing timing of gonadectomy (GDX) or sham (Sham) surgery, ERβ ligand (ERβ L) or vehicle (V) treatment, MWM cognitive assessment, and neuropathology (PATH). **b** Gonadally intact vehicle-treated midlife females (Sham + V, solid), gonadectomized vehicle-treated midlife females (GDX + V, diagonal lines) or gonadectomized ERβ ligand-treated midlife females (GDX + ERβ L, crossed lines) were assessed for spatial reference memory by MWM. Gonadally intact vehicle-treated mice showed preference for the TQ over the OQ ($p < 0.0001$), while gonadectomized vehicle-treated did not. In contrast, gonadectomized ERβ ligand-treated showed preference for the TQ over the OQ ($p = 0.007$). Blue line indicates the null hypothesis (25% in TQ). Sham + V, $n = 10$; GDX + V, $n = 6$; GDX + ERβ L, $n = 7$. Quantitative analysis of (**c**) LCN2+GFAP+ area fraction for astrocyte reactivity, (**d**) MHCII+IBA1+ area fraction for activated

microglia, (**e**) CLEC7A+P2RY12+ area fraction for disease-associated microglia (DAM), and (**f**) SYN1+PSD95+ area fraction for synaptic loss, each in dorsal hippocampus CA1 region in sham + V (solid), GDX + V (diagonal lines), and GDX + ERβ L (crossed lines). GDX + V showed increased reactive astrocytes ($p < 0.0001$), activated microglia ($p = 0.0003$), and disease-associated microglia (DAM) ($p = 0.0027$), as well as synaptic loss ($p = 0.0034$), each as compared to the Sham + V group. In contrast, there was a decrease in reactive astrocytes ($p = 0.0046$), activated microglia ($p = 0.0007$), and disease-associated microglia (DAM) ($p = 0.0346$), with increased synaptic staining ($p = 0.0445$) in GDX + ERβ L treated compared to GDX + V treated. Sham + V, $n = 8$; GDX + V, $n = 14$; GDX + ERβ L, $n = 4$. Two-sided Mann−Whitney $U$ test. Box plots with center lines showing the medians, boxes indicating the interquartile range, and whiskers indicating from the minimum and to the maximum values.

---

In summary, preclinical findings here showing beneficial effects of ERβ ligation in astrocytes of dorsal hippocampus in otherwise healthy midlife female mice suggest future studies targeting ERβ to maximize neuroprotective efficacy while minimizing general toxicity conferred by ERα. While it is not possible to target only astrocytes in humans, an estrogen receptor-specific and brain region-specific approach is possible. Treatments can preferentially target ERβ, and trial outcomes could focus on neurodegeneration in posterior hippocampus and prefrontal cortex. Together, this would build on past insights regarding the importance of estrogen type, dose, and timing to explore treatments for cognitive issues in menopausal women at midlife, an unmet need in half the population.

## Methods
### Animals and breeding strategy
All mice used in this study were from the C57BL/6J background. Mice were evaluated at three ages: young (3–4 months), midlife (12–14 months), and old (20–22 months). We generated mice with selective deletion of ERβ in astrocytes. GFAP-Cre:ERβ fl/fl (astrocyte ERβ

cKO) were generated by crossing the GFAP-Cre line (B6.Cg-Tg(Gfap-Cre)77.6Mvs/2J, JAX) with the exon 3 ERβ fl/fl line[57]. NSE-Cre:ERβ fl/fl (neuron ERβ cKO) were also generated by crossing the rNSEII-Cre line with the exon 3 ERβ fl/fl line[57]. mGFAP-Cre:RiboTag mice were generated by crossing RiboTag mice[96] with the GFAP-Cre line (B6.Cg-Tg(Gfap-Cre)77.6Mvs/2J, Jackson lab). To generate GFAP-Cre:RiboTag: ERβ fl/fl mice, RiboTag mice (B6J.129(Cg)-Rpl22 tm1.1Psam/SjJ, Jackson Lab) was crossed with GFAP-Cre: ERβ fl/fl.

In breeding of our mouse lines, Cre recombinase alleles were always inherited from females (mother), not from males (father). This is because one Cre line showed ectopic expression of Cre recombinase in the male germline (mGFAP-Cre 77.6 line; https://www.jax.org/strain/024098). Thus, paternal inheritance of Cre recombinase should be avoided. To this end, our breeding pairs were: (1) GFAP-Cre:ERβ fl/fl females with ERβ fl/fl males (to generate astrocyte ERβ cKO mice), (2) NSE-Cre:ERβ fl/fl females with ERβ fl/fl males (to generate neuron ERβ cKO mice), (3) GFAP-Cre:RiboTag females with RiboTag males (to generate astrocyte RiboTag mice), and (4) GFAP-Cre:RiboTag:ERβ fl/fl females with RiboTag: ERβ fl/fl males (to generate astrocyte ERβ cKO RiboTag mice).

Genotypes were assessed by PCR using the following primer sequences (5′-3′): ERβ, CTTCTTAGAGGTACGGATCCC and AATCTCTT TGCCTTCCAGAGC; and Cre, GCACTGATTTCGACCAGGTT and GCTA ACCAGCGTTTTCGTTC; *LoxP-STOP-loxP-Rpl22-HA* (RiboTag), GGGA GGCTTGCTGGATATG and TTTCCAGACACAGGCTAAGTACAC. All mice were housed in a facility with 12-h light/dark cycles with temperature and humidity controlled and were allowed free access to food and water. All procedures were done in accordance with the guidelines of the National Institutes of Health and the Chancellor's Animal Research Committee of the University of California, Los Angeles Office for the Protection of Research Subjects. All animal experiments were approved by the Chancellor's Animal Research Committee of the University of California, Los Angeles Office (ARC#1998-001).

### Gonadectomy
Gonadectomy was performed in female mice at 2 months of age. Before the procedure, carprofen (1.2 mg/ml per animal) and saline (0.5 ml) were injected subcutaneously. Mice were anesthetized by inhalational anesthesia with isoflurane. The fur above the lateral dorsal back was removed and the skin was sterilized with betadine and alcohol scrubs. Bilateral incisions were made into skin and the ovaries were removed. The muscle layer was closed with absorbable suture and the skin layer was closed with wound clips. For postoperative care, amoxicillin (50 mg/ml) was added to the water for 5 days for antibiotics (0.5 mg/ml water) and another dose of carprofen (0.1 ml) was given within 24 h of surgery. Wound clips were removed 7 days later.

### Treatment
For ERβ-ligand treatment we used diarylpropionitrile (DPN, R&D). Treatment was initiated 1 week after gonadectomy. Briefly, DPN was dissolved in 100% ethanol and then mixed into 100% Miglyol®812 (CREMER Oleo GmbH & Co. KG) at 1:9 ratio making the final EtOH concentration 10%, and given subcutaneously every other day at a dose of 8 mg/kg/day until the end of each experiment. All assessments were done in a blinded fashion with regards to knowledge of treatment randomization.

### Behavioral testing
Behavioral tests were performed during the light cycle. All the experimental mice were transferred to the behavior testing room at least 30 min before the tests to acclimate to the environment. Temperature and humidity of the experimental rooms were kept at $23 \pm 2\,°C$ and $55 \pm 5\%$, respectively. The brightness of the experimental room was kept dim. All animals were habituated to handling for 1–2 min every day for at least a week prior to experimentation. All experimental procedures were in accordance with the Animal Research Committee at the University of California, Los Angeles. The experiments were conducted in the UCLA Behavioral Testing Core Facility and the investigators who performed experiments and analyses were blinded to mouse group.

### Morris Water Maze (MWM)
Mice were trained in hidden platform MWM task to investigate spatial learning and reference memory[97,98]. In the hidden-platform test, the animal learns the location of the escape platform in relation to the cues in the periphery of the room, the escape platform remains in the same location (in the center of NW, named as Target Quadrant (TQ)) for all training trials and is located 1 cm below the level of water made opaque with non-toxic white tempera paint. Pool temperature was maintained at $23–25\,°C$ by the addition of warm water. On the first day of training, mice are placed on the platform for 30 s before the first trial. If the mouse jumps from the platform, they are returned to the platform for a cumulative 30 s. After 2 min delay, the first day of training sessions begins. Mice undergo 3 training trials a day for 5 days. In each training trial, mice

are placed in the pool facing to the edge, 5 cm from the wall in the center of one of the three pool quadrants not containing the escape platform, and the latency to finding the platform is recorded. Each trial ends either when the mouse successfully found the platform or after 1 min. At the end of each trial, mice are allowed to remain on the platform or placed on the platform for 5 s before returning to their home cages. On day 5, 2 h after training, the platform is removed for the probe trial. Mice are recorded for 60 s and assessed for how long they spend in each quadrant searching for the platform. All recordings of probe trial are analyzed using TopScan (Clever Sys, Inc., Reston, VA).

### Y maze
Mice were trained on the Y maze to evaluate working memory on a continuous spontaneous alternation task. The apparatus of Y maze is constructed from clear plexiglass to allow for visualization of external spatial information. Mice are placed in one arm of a Y maze, and spontaneous alternation is recorded in a single continuous 5-min trial by 2–3 live observers. Each of the three arms is designated a number 1–3, and entries into the arms are recorded (i.e., 1, 2, 3, 2, 1). The observers are blinded to the experimental groups recorded arm entries. Spontaneous alternation is calculated as: (number of alternations / (total arm entries − 2)) * 100. The apparatus is cleaned with 50% Windex, between trials.

### Magnetic resonance imaging (MRI)
All animals were scanned in vivo at the Ahmanson-Lovelace Brain Mapping Center at UCLA on a 7T Bruker imaging spectrometer with a micro-imaging gradient insert with a maximum gradient strength of 100 G/cm (Bruker Instruments, Billerica, MA). An actively decoupled quadrature surface coil array was used for signal reception and a 72-mm birdcage coil was used for transmission. For image acquisition, mice were anesthetized with isoflurane and their heads secured with bite and ear bars. Respiration rate was monitored and the mice were maintained at $37\,°C$ using a circulating water pump. Each animal was scanned using a rapid-acquisition with relaxation enhancement (RARE) sequence with the following parameters: TR/TEeff 3500/32 ms, ETL 16, matrix: $256 \times 192 \times 100$, voxel dimensions: $100 \times 100 \times 100\ \mu m^3$. Total imaging time was 93 min. Images were acquired and reconstructed using ParaVision 5.1 software.

### Atlas-based morphometry (ABM)
All MRI brain images were skull stripped using BrainSuite 19b[99] (http://brainsuite.org). Images were then processed and examined with Statistical Parametric Mapping (SPM) 8 software (Welcome Trust Center for Neuroimaging, London, United Kingdom; http://www.fil.ion.ucl.ac.uk/spm) and SPMMouse (SPMMouse, http://www.spmmouse.org)[100] within MATLAB version 2013a (MathWorks, Natick, MA). Images were manually registered to tissue probability maps (TPMs) generated from 60 C57BL/6J mice[28] using 6-parameter linear transformations. The images were bias corrected and segmented into GM, WM, and cerebrospinal fluid (CSF) using the unified segmentation algorithm[101]. The resulting segments were used to create a Diffeomorphic Anatomical Registration using Exponentiated Lie algebra (DARTEL) template[102]. Bias-corrected images were warped to the DARTEL template, and averaged to create a mean template for visualization. Brain region labels were drawn onto the mean template and then inverse warped back to the individual images where they were used to measure region volumes. Analysis on atlas-based morphometry results was done using R (R Core Team, 2022, http://www.R-project.org/).

### Histological analysis
Standard histological analysis methods were applied as we described previously[57,59,80,103–105] with some modifications.

## Tissue preparation

Mice were deeply anesthetized in isoflurane and perfused transcardially with ice-cold 1× PBS for 5–7 min, followed by 4% paraformaldehyde for 5 to 7 min. CNS tissues were dissected and submerged in 4% paraformaldehyde overnight at 4 °C, followed by 30% sucrose for >24 h. Brains were embedded in 7.5% gelatin/15% sucrose solution. The gelatin-embedded tissues were submerged in 4% paraformaldehyde overnight at 4 °C, followed by 30% sucrose for >24 h, and stored at −80 °C after flash frozen by dry ice. The 40-µm thick free-floating brain sagittal-sections were prepared using a cryostat (Leica Biosystems) at −20 °C. Tissues were collected serially and stored in 1x PBS with 0.1% sodium azide in 4 °C.

## Immunohistochemistry

Prior to histological staining, 40-µm thick brain free-floating sections were thoroughly washed with 1X PBS to remove residual sodium azide. Tissues were incubated with 50% methanol/PBS (1:1) at RT, followed by 1xPBS washing. After heat antigen retrieval with heated 10 mM Citric Acid−0.05% Tween 20, pH6.0, all tissue sections were permeabilized with 0.3% Triton-X and 2% normal serum in 1X PBS for 10 min on ice and blocked with 10% normal serum in 1X PBST for 1 h. Tissues were then incubated with primary antibodies in 1xPBST-2% normal goat serum overnight at 4 °C. The next day, tissues were washed and incubated with secondary antibodies in 1xPBST-2% normal serum for 2 h at RT. Sections were mounted on slides, allowed to dry, and cover slipped in fluoromount G (Southern Biotech) for confocal microscopy. The following primary antibodies were used: Goat anti-mouse Lipocalin-2 (LCN2) Antibody (at 1:50, R&D, cat#AF1857), Rat anti-GFAP (at 1:500, Thermo Fisher, cat# 13-0300, clone 2.2B10), Rabbit anti-Iba1 (at 1:1000, Wako Chemicals USA Inc. cat# 019-19741), Rat anti-MHCII (at 1:500, Biolegend, cat#107602, clone M5/114.15.2), Rabbit anti-P2RY12 (at 1:1000, AnaSpec, cat#AS-55043A), Rat anti-CLEC7A (at 1:200, InvivoGen, cat#mabg-mdect, clone R1-8g7), Rat anti-P2RY12 (at 1:100, Biolegend, cat#848002, clone S16007D), Rabbit anti-CLEC7A (at 1:200, Thermo Fisher, cat#PA-534382), Rabbit anti-Synapsin 1 (at 1:500 dilution, Synaptic Systems, cat#106 103), Guinea pig anti-PSD95 (at 1:250 dilution, Synaptic Systems, cat#124 014), Mouse anti-HA (at 1:500, Biolegend, cat#901522, clone 16B12), Rabbit anti-Mouse ALDH1L1 (at 1:250, Abcam, cat#ab87117), Mouse anti-CC1 (at 1:500, Millipore, cat#OP80, clone CC-1), Rabbit anti-mouse ERβ (at 1:100, Thermo Fisher, cat#PA1310B), Guinea pig anti-NeuN (at 1:500 dilution, Synaptic Systems, cat#266 004). The following secondary antibodies were used at 1:500 dilution for staining the tissues: Goat anti-rabbit-Alexa Fluor Plus 647 (cat#A32733, Thermo Fisher), goat anti-Guinea Pig - DyLight 550 (cat# SA5-10095, Thermo Fisher), goat anti-rat-Alexa Fluor488 (Cat# 112-545-167, Jackson Immunoresearch), and goat anti-rat-Cy3 (cat#112-165-167, Jackson Immunoresearch), Goat anti-rat Cy5 (cat# 112-175-167, Jackson Immunoresearch), Goat anti-rabbit TRITC (cat# 111-025-144, Jackson Immunoresearch), Donkey anti-goat Cy5 (Ab#6566, Abcam), Donkey Anti-rat Cy3 (cat#712-165-153, Jackson Immunoresearch), Goat anti-mouse- Cy3 (cat#115-545-166, Jackson Immunoresearch).

## Confocal microscopy and image analysis

Stained sections were examined and imaged using Olympus BX61 DSU fluorescence microscope with a Hamamatsu ORCA-FLASH4.0LT + SCMOS CAMERA. Images were taken in stacks. All images were taken and processed using the integrated software program Olympus *cellSens* Software Version 4. ImageJ (NIH) was used to perform integration and analysis of images.

## RNA co-immunoprecipitation, sequencing, and analysis

Mice were exposed to a lethal dose of isoflurane and transcardially perfused with ice-cold 1X PBS for 3–5 min, followed by with ice-cold 1% paraformaldehyde-1X PBS for 4–5 min. Hippocampus were collected and snap-frozen in liquid nitrogen. Tissues were stored at −80 °C. Frozen tissues were homogenized on ice using Dounce Homogenizer with freshly made homogenization buffer containing: 50 mM Tris-HCl (Invitrogen) pH 7.5, 100 mM KCl (Fisher scientific), 12 mM MgCl₂ (Fisher Scientific), 1% Nonidet P-40 (Roche), 1 mM DTT (Sigma-Aldrich), 1x Proteinase Inhibitors Cocktail (Sigma-Aldrich), 200 units/ml RNAsin (Promega), 100 ug/ml cycloheximide (Sigma-Aldrich), and 1 mg/ml heparin (Sigma-Aldrich). The homogenates were then centrifuged at 20,000 × *g*, 4 °C for 15 min. The supernatant was collected and incubated with pre-washed anti-HA conjugated magnetic beads (Pierce) overnight on a rotating wheel at 4 °C. After removal of the supernatant, the magnetic beads were washed 3 times with high salt buffer containing: 50 mM Tris (pH 7.5), 300 mM KCl, 12 mM MgCl₂, 1% Nonidet P-40, 1 mM DTT, 100 µg/ml cycloheximide. Then, 25 µl of proteinase K (4 mg/ml; Zymo Research) was added to the samples and incubated in a 55 °C water bath for 30 min. After incubation, 300 µl Tri-reagent was added, and the Direct-zol™ RNA MiniPrep Plus kit (Zymo Research) was used for RNA isolation according to manufacturer's instructions. RNA quantity and quality were measured using NanoDrop 2000 spectrophotometer (Thermo Scientific) and Agilent High Sensitivity RNA ScreenTape System (Agilent).

The RNA sequencing library was made using KAPA Stranded RNA-Seq Kit (Kapa Biosystems) which consists of mRNA enrichment, cDNA generation, end repair, A-tailing, adapter ligation, strand selection, and PCR amplification. Barcoded adapters were used for multiplexing samples in one lane. Sequencing was performed on Illumina HiSeq3000 for a single end 1 × 50 run. Data quality check was done on Illumina SAV. De-multiplexing was performed with the Illumina Bcl2fastq2 v 2.17 program. These procedures were performed at the UCLA Technology Center for Genomics and Bioinformatics core facility.

Sequencing analyses and production of figures were performed in R (R Core Team, 2022, http://www.R-project.org/). Qualities of raw sequence data were examined using FastQC (http://www.bioinformatics.babraham.ac.uk/projects/fastqc/), and Trimmomatic[106] was used for cleaning. R package "QuasR"[107] was used for the read alignment to the mouse genome (mm10) followed by counting at the gene level. The genes with count numbers less than 1 in more than 3 samples were filtered out. Differentially expressed genes between Sham WT and GDX WT as well as between Sham WT and Sham ERβ cKO mice were identified with R package "edgeR"[108]. False discovery rate of 0.1 was used as the threshold of differentially expressed genes. Heatmaps were created using the R package "gplots". Canonical pathway enrichment analysis was performed for differentially expressed genes using Ingenuity Pathway Analysis (QIAGEN, Redwood City, www.qiagen.com/ingenuity).

## Human microarray analysis

Human hippocampus microarray data for normal brain aging was obtained from the GEO database (GSE11882, Ages: 26–99 years old). The genes with hybridization intensity above 100 in more than 50% of arrays were used for further analysis. Log2 transformed intensity values were quantile normalized using the "affy" package in R (R Core Team, 2022, http://www.R-project.org/). For the genes with multiple microarray probes, mean values were used. Correlation between gene expression level and ages was assessed by Pearson correlation analysis. Gene expression difference was examined using two-sided Mann–Whitney *U* test.

## Statistical analysis

Mice were randomly assigned to experimental time points or groups. Investigators were blinded to group allocation during data collection and analysis. Points represent individual animals. All box plots with center lines showing the medians, boxes indicating the interquartile range, and whiskers indicating either a maximum of 1.5 times the

interquartile range beyond the box or from the minimum and to the maximum values. Prism 9 (GraphPad) was used for all behavioral tests and histological data analysis. The statistical difference between groups was determined using two-sided Mann–Whitney $U$ test. Independent factor effect and interaction between two factors was calculated by two-way ANOVA. Correlation was assessed by Pearson correlation analysis. The exact $p$ values ($p \leq 0.05$) were shown on each graph.

All statistical analysis for MRI data was conducted using R (R Core Team, 2022, http://www.R-project.org/). MRI data were first analyzed using a two-way ANOVA and then was followed up with two-sided Welch's $t$-tests. Regression analyses are reported as Pearson correlation coefficients.

### Reporting summary

Further information on research design is available in the Nature Portfolio Reporting Summary linked to this article.

### Data availability

The mouse astrocyte RNA-Seq data generated in this study have been deposited in the GEO database under accession code GSE220288. Source data are provided with this paper.

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

## Acknowledgements

We thank members of the Voskuhl laboratory Heaveen Ahdi, Valerie Vessels, Vincent Tse, and Ellis Jang for technical assistance. We thank UCLA Behavioral Testing Core Facility, especially Dr. Lindsay M Lueptow and Ms. Irina Zhuravka for consultation. This work was supported by the U.S. National Institutes of Health (NIH) (R35NS132150, R01NS096748 and R01NS109670 to R.R.V., R21NS121806 to A.M.G., and 1F31NS105387 to C.E.M.); the Conrad N. Hilton Foundation (#17734 and #18394 to R.R.V.); the Nancy Davis Race to Erase MS Foundation, and the Dunk MS Foundation.

## Author contributions

R.R.V. conceived the study, supervised the team, provided oversight and generated funding. N.I., Y.I., C.E.M., A.M.G., and R.R.V. designed the experiments and wrote the manuscript. N.I. prepared all animals, tissues, and transcriptomes for this study, and performed data analysis for behavioral studies. C.E.M. and A.M.G. performed MR imaging and the data analysis. N.I., T.T.S., and D.C.D. performed behavioral tests. N.I., T.T.S., D.C.D., M.R.L., S.W., and S.S.S. performed immunostaining and the analysis. L.C.G. and N.I. performed HA/ER β expression study. Y.I. analyzed RNA sequencing data and human GEO data. All authors discussed the data.

## Competing interests

The authors had full access to all of the data in this study and take complete responsibility for the integrity of the data and the accuracy of the data analysis. R.R.V. is an inventor on UCLA's patents pertaining to estriol and is a consultant with equity in CleopatraRX, the licensee. R.R.V. is also an inventor on UCLA's patents pertaining to novel estrogen receptor beta ligands which are unlicensed. Other authors have nothing to disclose.
