## [Peer Review File · Nature Communications]

Estrogen receptor beta in astrocytes modulates cognitive function in mid-age female miceReviewers' comments:

Reviewer #1 (Remarks to the Author):

A. Key findings:

Itoh et al. seek to refine hormone replacement therapy (HRT) by gaining insight into brain areas affected by hormone removal, as well as the distinct roles of estrogen receptors (alpha vs beta) and of neurons vs astrocytes. To this end, behavioral and structural analyses of brain areas with magnetic resonance imaging (MRI) are performed in three studies. One is a longitudinal analysis of male and female mice throughout life. Another study explores the effects of gonadectomy at midlife. And a third study examines mice with selective removal of estrogen receptors in astrocytes or neurons. They report age-, sex- and brain region-dependent atrophy that is exacerbated by gonadectomy and mimicked by targeted deletion of estrogen receptor beta in astrocytes. Taken together, the mouse data suggest that the neuroprotective actions of estrogens are cell- and receptor-dependent, primarily affecting dorsal hippocampus. It is also remarkable that gonadectomy results in severe decrease of spines in dorsal hippocampus. The study may inform structural analyses in HRT, as well as therapeutic strategies, with a focus on the emerging avenue of astrocyte-targeted therapies. The authors are to be commended by the hard work and the wealth of tests and animal models. Tests include hippocampal-dependent spatial memory (Morris Water Maze, MWM), working memory (Y-maze) and contextual fear conditioning. Of great value is the use of MRI to dissect out regional changes in association with behavioral deficits. Mouse models include intact and gonadectomized, as well as knock out mice for estrogen receptor types in astrocytes and neurons.

B. Limitations:

B.1. Insufficient clarity.

Conceptual: The gap of knowledge ('understanding brain region-specific, cell-specific and receptor-specific mechanisms') that the authors aim to dissipate should be rephrased. As I understand it, the authors aim to clarify why HRT does not mitigate cognitive deficits and brain atrophy in menopausal women (problem statement) and propose that the conflicting results may arise from paradoxical actions of different estrogen receptors in different cell types, and the fact that readout measures are not directed to the right brain areas. There is a confusion in the said gap between actors (cells and receptors) and effects (brain areas and cognition), for, in the study, brain regions are not specifically targeted. The gene knockout affects cells all over the brain, and clinical therapies are systemically administered. The repeated expression 'region-specific targeting' throughout the text should be eliminated or clarified.

Experimental: Perhaps it is just me, but the study rationale is not clear, particularly which tests are used, and which ages analyzed. Specifically, in figure 1, in intact mice only MWM is performed at three ages, with negative results, while in Fig 2 (gonadectomized mice) Y Maze and contextual fear conditioning are incorporated, but only in two ages. The comparison of young and midlife mice allows to test the interaction between age and absence of hormones, but the absence of old mice makes it difficult to conclude whether gonadectomy at midlife accelerates cognitive and structural decline to match that of old ages. Also, Fig. 2a and Fig. 2c repeat data. Perhaps Fig. 1 and Fig. 2 should be condensed into a single figure to present the model of intact (normal scenario) and gonadectomized mice (extreme scenario) and conclude which behavioral and structural analyses are more apt to study the effect of hormones. Likewise, in Fig. 4 (studies in ER knockout mice) Y Maze is not used, only midlife mice are tested, and immunohistochemical analyses of GFAP, Iba1 and PSD95 are not performed.

Fig. 3 e, f. I guess that it is normal that '% time in other quadrants' is the mirror/inverse version of '% time in target quadrant'. If so, why is the scale in the Y axis of Fig. 3f different from the one in 3e? For example, a 20% time in target quadrant should correlate with 80% time in other quadrants.

B.2. Overinterpretation of correlations.

Direct results should be differentiated from interpretations. For example, line 212 'Neuroprotection in females at midlife is mediated by ERbeta in astrocytes' is not a correct label. The experimental design will show the effect of ERbeta removal, but it does not demonstrate that the deleterious mechanisms are the same as those mediating damage caused by gonadectomy, which affects many hormones. Such an equivalence is a correlation. Reversal experiments such as addition of ERbeta agonists to gonadectomized mice may beautifully strengthen the connection the authors wish to make.

C. Methodologies:

Supplemental Fig. 1. presents data from estrogen receptor alpha knockout mice that are not described in Methods. These data are important and could be moved to the main text. Likewise, the PCR results confirming successful deletion of estrogen receptor genes should be shown.

It would be nice to have images of MRI in different experimental scenarios in Fig. 1.

The quantification of immunohistochemical data is not explained in sufficient detail, including the number and distance between sections, stereological considerations, programs used to quantify, thresholding, background subtraction, etc. Also, images of PSD95 immunostaining, perhaps the extreme cases, would be useful to visually support one of the most interesting findings of the study.

D. Style issues:

The authors may consider removing 'Mind the Gap' from the title, since journalistic expressions should be avoided from scientific reports, and it is not clear which gap should be minded until one reads the introduction. Also, the title should describe the model and main result of the study and not the implications thereof, as noted. Something like 'Selective deletion of ERbeta in astrocytes but not neurons mimics behavioral and structural deficits caused by gonadectomy in female mice at midlife' may work.

The manuscript is well written, but there is always room for improvement. In the Abstract, the sentence in line 26 starting with 'Some aspects of aging...' is dispensable. I already suggested rephrasing of the problem statement above, in noting that the results should be described in a neutral manner, distinguishing findings from interpretations. For instance, in line 30, an alternative to the current sentence is: Gonadectomy impairs behavioral performance in *** test (specify) and results in atrophy in *** but not in **** (specify) in **** mice (specify), suggesting that ovarian hormones in **** mice (specify) protect against hippocampal-dependent cognitive impairment and dorsal hippocampal atrophy. Likewise, in line 33: Deletion of ERbeta in astrocytes but not neurons by **** technology (specify) in **** mice (specify) result in **** (specify), as measured by **** (specify), suggesting that the protective effects of estrogen in midlife female mice are mediated by ERbeta receptors in astrocytes.

Line 52 'during health' is unnecessary. The study does not need to touch on the controversy of whether there is healthy or unhealthy aging, or aging is always unhealthy. Also, the Marxian sentence (from Groucho) 'A better understanding of the effect of aging can provide insights into the effect of aging' should be rephrased for clarity.

Line 59. It appears from the sentence that the impact of timing and estrogen type (does it mean receptor type?) has been already clarified according to the literature (refs 6-9), in contradiction with the problem statement, and with line 88 (refs 8, 28). Please, succinctly specify what is known and what are the areas of conflict.

Line 62. This sentence belongs to a review, there is no need to defend the need for separation of sexes in a study using, moreover, a reference from 2012. Likewise, the rest of the paragraphs can be summarized, or moved to the Discussion. For example, it should be highlighted in the Discussion that the mice mirror the sex-dependent evolution of cognition and atrophy in humans.

Line 91. Again, the study does not inform about brain region-specific therapeutic approaches.

What it does is to unravel which brain regions are more affected by the experimental manipulations; nor the authors address the regional heterogeneity of astrocytes or neurons, for their genetic manipulations are global.

Lines 212 and 254. These two paragraphs can be condensed into one. Also, the explanation of the contextual fear conditioning (line 256) should be moved to earlier in the text, when the meaning of the different tests is explained.

In Discussion, the clinical implications of the findings can be summarized. Text condensation is recommended.

As most articles, including those written by English-speaking authors, the manuscript may benefit from professional copy editing.

Reviewer #2 (Remarks to the Author):

In this study, Itoh, Meyer, et al. studied the effect of sexual hormone loss in age-related decline of memory, and the involvement of Estrogen Receptor β (ER β) in mediating the aggravated hippocampal neuropathy observed in female mice following gonadectomy. The work combines memory assessments, MRI imaging of structures important for memory formation, and immunohistochemistry. The paper is clear, and the methodology thorough, with appropriate conclusions when describing the result. However, I believe the claims to be overstated regarding the potential of astrocytic ER β as neuroprotector, based solely on the presented data, without taking into account the existing literature. The work nicely shows that the deletion of Er β is detrimental for cognition from midlife, but in my opinion, the authors have not thoroughly demonstrated the beneficial effect of ER β activation on cognitive aging. As I elaborate below, I would also advise the authors to highlight the novelty of their finding, and finally to be cautious regarding the interpretation of their research, done in the context of normal aging but not neurodegeneration, as stated several times throughout the text.

Title: While I understand the underlying idea, i.e. that ER β in astrocytes could be an important candidate for future therapeutic targets in cognitive aging in female. I tend to find it overclaiming. The authors do not demonstrate that ER β solely mediates the potential improvement of memorisation in aging, for example using a specific drug targeting those receptors, or losing a positive effect after the deletion.

1. Behaviour: I was very surprised to read that the authors did not find an age-related defect of spatial memory in the Morris Water Maze (MWM) experiment (Fig1). It is a well-established fact that hippocampus-dependent types of memory, such as spatial, reference, or episodic memory, are sensitive to aging across species (please see the body of work from Catherine Barnes, Howard Eichenbaum, Michela Gallagher, Michael Yassa, Aline Marighetto, Sara Burke, to name a few). In trying to understand this lack of defects, I came to think that the paper would benefit not only from showing the performance during acquisition, at least in the supplementary material, as a control of learning, but also from providing more details about the paradigm. Indeed, depending on the protocol, the memory can go from spatial to procedural (i.e., hippocampus-independent), which remains relatively intact during aging. Alternatively, the authors could also represent the target quadrant exploration during the test as blocks of 15 sec to clarify the mouse behaviour. That being said, I think the high performance from aged mice is due to the fact that the authors tested the mice 2h after the last training session. This could explain the difference between retrieval in the MWM and the fear conditioning, which was tested 24h post-conditioning, and did reveal the age-related memory decline. I would suggest to the authors to homogenise the conditions of behavioural testing throughout the paper, for clarity of understanding and comparison, or at least to address these discrepancies in the discussion. As the age-related decline of hippocampus-dependent memory is such a long-standing fact, I don't think the study benefits from starting with such a clashing result and could deter the readership. Especially since in GDX mice, the defect appears earlier, at "mid-life", and could be interpreted as early aging. Finally, regarding the behaviour part, I would like to draw the authors' attention to the fact that

the dichotomy in the role of the hippocampus to sustain recent but not remote memory has been revised in recent years (Goshen et al., Cell 2011) benefiting from the advancement of optogenetics. While I agree with the authors' conclusion regarding their own result, I would advice to modify the introduction of this part to take into account recent literature rather than Frankland and Bontempi's early work. A minor remark on this part, I am sure what is the authors' rationale for making this data as a supplementary figure only, when this is an interesting result.

2. ER β cKO. I think it would be helpful to provide more details regarding the origin of the mice and selectivity of the conditional KO. Can you provide a proof of the specificity of the deletion within the astrocyte or the neurons? Could there be a compensation of the Era, that would underlie the result and hinder the interpretation regarding the role of ER β ? Providing an image illustrating the specific deletion in both cell type is, in my opinion, a major control to ensure the validity of study.

3. Neurodegeneration: One major point that I would like to draw the authors' attention towards, is their multiple reference to neurodegeneration throughout the paper, and the lack of distinction between normal aging and pathological aging. From what I understand, the authors seem to suggest that their study addresses neurodegeneration. However, the study has been conducting in C57Bl/6j mice, which do not develop pathological aging (i.e., dementia, Alzheimer's-like memory defects), as demonstrated in my previous point, these wildtype aged mice can demonstrate high performance in certain conditions (Fig1), which would not be the case in pathological aging. In normal aging, the memory defects are associated with atrophy, but not neuronal loss, unlike pathological aging. Moreover, the underlying mechanisms, and progression of alterations are widely distinct according to the type of aging. Therefore, I do not think it is proper in their condition to refer to neurodegeneration and would reframe the paper as normal aging. They could address the potential importance of their work regarding pathological aging in the discussion, but the use of wildtype mice do not support the direct extrapolation towards neurodegeneration.

4. Discussion: I think the work would benefit from stressing out throughout the text and in particular the discussion, the novelty of the research, and replacing the work within the literature. It is my belief that aside from the very interesting distinctive role of ER β in astrocytes vs. neurons, the rest of the work has in some form been already reported. Age-related memory deficits (see authors in part 1 for reference); There is also a large body of work on the sexual dimorphism in cognitive aging (e.g. Frick et al., 2000; for review, Frick et al 2008), and on the role of estrogens in the brain and learning and memory, and effect of ovariectomy, which deserve to be cited in my opinion (L. Galea, K. Frick, E. Waters, M. Adams, C. Wolley, B. McEwen – non-exhaustive list). Similarly, the atrophy of the hippocampus has been demonstrated multiple times (for review Barnes and Burke, 2010). Finally, the reduction in spine density following ovariectomy has been previously reported by the McEwen laboratory (Gould et al., 1990). I do not intend to denigrate the study; I simply believe that the authors would benefit from valorising their findings in light of the existing literature.

Final point on the discussion, I fail to understand why the authors address hormone replacement and MS in such length, when their study does not use estrogen supplementation, potential effect on alteration similar to MS, and that MS have limited cognitive effect. Rather, I would be very interested in reading the authors' take on the integration of their findings about sexual dimorphism, from cognition to synaptic loss and the specific role of glial cells in these processes.

Minor comment: I think the reader would benefit from the homogenisation of the statistical report, which at the moment is alternatively in the legend or the text.

Looking forward to reading your response,

AS Al Abed

Reviewer #3 (Remarks to the Author):

This article provides original and interesting data regarding the neuroprotective role of the astroglial estrogen receptor (ER) beta against hippocampal-dependent cognitive deficits and

atrophy in menopausal women. The authors combine molecular approach (constitutive conditional knockout mice) with in vivo MRI and behavioral testing to assess the role of astroglial ER beta in protecting against brain atrophy (in vivo MRI) and cognitive deficits, assessed via spatial reference memory (Morris Water Maze), working memory (Y-maze) and contextual learning (contextual fear conditioning). The data are primarily correlative and descriptive. By using a conditional KO mice, the authors conclude that ER beta in astrocytes could be a novel promising therapeutic target for cognitive deficits and hippocampal atrophy in menopausal women. This is of interest. Yet the conclusions are not well supported by the experimental data mostly because of the molecular approach, which is not appropriate. In addition, the manuscript lacks mechanistic insights. Thus although interesting, this work is preliminary in its present form and presents several major issues.

1) The authors use GFAP-cre mice to constitutively delete ER beta in astrocytes. These mice cannot be used to conclude about implication of astroglial proteins, when these proteins are also expressed in neurons.

There are indeed major caveats in using the GFAP-Cre mice, as it has been well documented by numerous groups that deletion occurs in both neurons and astrocytes, since GFAP is expressed in precursor cells during early development. This confounds the authors major conclusion that deletion of ER beta from astrocytes contributes to cognitive deficits and brain atrophy in menopausal female.

A few papers here listed well describe these caveats :

"Looks Can Be Deceiving: Reconsidering the Evidence for Gliotransmission"

<https://doi.org/10.1016/j.neuron.2014.12.003>

Germ-Line Recombination Activity of the Widely Used hGFAP-Cre and Nestin-Cre Transgenes

<https://www.ncbi.nlm.nih.gov/pmc/articles/PMC3857304/>

Expression Specificity of GFAP Transgenes

<https://link.springer.com/article/10.1007/s11064-004-6881-1>

Other experiments should thus be combined with the use of these mice, such as the use of inducible cKO mice and/or rescue experiments targeted to astrocytes (re-expression of ER beta specifically in astrocytes in the astroglial cKO mice).

2) There is no validation of these newly generated astroglial- and neuronal-specific ER beta transgenic models. One would like to see quantification of ER beta expression in astrocytes and neurons in both cKO mice (gfap-cre:ER beta^{fl/fl} and nestin-cre:ER beta^{fl/fl}).

3) There is a lack of details regarding the mice (origin (stock number or laboratory source), breeding strategy controlling germline recombination, appropriate littermate controls. This precludes evaluation of whether the authors have used appropriate controls.

4) The authors use GFAP immunostaining to assess astroglial reactivity. GFAP expression can vary independently of astroglial reactivity. The authors should thus use additional markers (such as vimentin, stat3, GS...) and analyse astrocyte morphology to conclude about astrocyte reactivity.

5) The authors report synapse loss in menopausal female. They however only performed PSD95 staining to assess synapses. A synapse is composed of a pre- and a postsynaptic element and can only be identified by the colocalization of both elements. Staining for both pre- and postsynaptic markers should thus be performed to quantify synapse number.

6) This manuscript completely lacks mechanistic insights. How ER beta in astrocytes contributes to brain structure and cognitive performance in male and female? Does this require a crosstalk with other cell types such as microglia?

Reviewer #4 (Remarks to the Author):

The paper "Mind the Gap: Estrogen receptor beta (ER β) in astrocytes is a therapeutic target to

prevent cognitive problems at menopause" investigates the impact of estrogen, and more specifically estrogen receptors beta, to the observed gender differences in behavior, hippocampus volume and astrogliosis in rodents at midlife. The authors use a combination of techniques like MRI, histology, behavior on wild type animals, animals which underwent gonadectomy/sham and ERb knock-out animals.

The narrative is very interesting, the results important and well contextualized. The work is extremely significant and timely. The methodology is rigorous, although a few important pieces of evidence are missing to have a more balanced picture and close the story (detailed below). I also have doubts on some of the statistical approaches chosen. The paper is easy to follow and very well written.

There are a few aspects that can be improved. Below, my suggestions.

1

Fig.1 : the information contained in a-b and c-d is redundant. In addition, doing twice a scatter plot, which distributes points random, on the same data generates plots which are visually different, which is awkward. The same applies to Fig. 2 a-b and c-d, and Fig. 4 a-b. I think half of the plots can safely go to supplementary.

2

Following the previous comment, I think the most correct statistical approach would be a single ANOVA for the water maze data where sex is a factor (in addition to age and quadrant), followed by appropriate post-hoc comparisons. The same applies for MRI volumetric data, where ROI should be another factor (cortex, striatum, HC).

3

In Fig 1 h, it seems that there is no volume change in HC in male mice across the lifespan. This results somehow weakens the point made by the authors in the rest of the paper, about the importance for neuroprotection across the lifespan of the ERb astrocyte receptors specifically in the dorsal hippocampus, unless the dHC is only important in ageing female, which is hard to digest, unless supported by evidence. Generally speaking, the paper is missing showing and discussing in males at least some of the analyses performed in females, for example Fig 2 g-i.

4

Fig 1, supplementary excel table: why using Welch test? Is it because the samples have unequal variance? Please specify in the methods.

5

Fig 1: It is noteworthy that from Fig. 1 there is no relation between structure and function, at least in the context of ageing. If it is a matter of different sensitivities of the two techniques employed (MRI volumetry vs behaviour), as hinted by the authors, I suggest they discuss the issue and report the power analysis used for determining the sample size. Perhaps the behavior is underpowered?

6

Fig.2: Similarly to comments 1 and 2, the figures of the water maze are redundant, and the statistical model could be just one for each test but include everything (gender, therapy, life stage).

7

Fig. 2 f: why freezing was only measured at midlife?

8

Fig. 2 j-k: I suggest to only plot regression line when significant. Also, there seems to be a lot of points in the plots. Are all animals (both sexes, all ages, GDX/sham) pulled together? If so, which is the rationale? Perhaps the different groups could be shown in the plot by coloring the datapoints differently.

9

Fig. 3, Neuropathology: the authors talk about "area fraction" but what is exactly measured and how? Can distinct aspect of the morphology be measured (ramifications density, cell body size, cell number etc.? This is important as the term "glia activation" can refer to distinct morphological changes happening to the cell. In general, the method->histology section is missing details needed for the reader to be able to reproduce the analyses.

10

Line 194: the notation is misleading. Perhaps by "direct" and "indirect" correlations the authors mean linear positive and linear negative? Or do the authors really mean indirect as mediated by something else? If so, specify.

11

Fig 3 is missing the plots of the correlations between time in TQ and gliosis (data is discussed but not shown).

12

Following the rationale of the paper, Fig 4 is missing the neuropathology. How are the astrocytes affected by ERb deletion? Are they more activated at midlife?

13

Following the previous comment but more generally, what is ultimately the mechanism that the authors identified as leading to atrophy and worst cognition in ageing females? Is it astrogliosis? If so, the authors should report more neuropathological data (or other equivalent astrogliosis markers) to make a stronger point.

14

It is interesting that there is no cognitive effect in young female mice which lacks endogenous hormones or receptors. What about HC volume?

15

As the authors point out in the discussion, microglia also have ERb. Can the author show and compare some histology of microglia in animals lacking endogenous receptors and animals lacking ERb in astrocytes?

Reviewers' comments:

Reviewer #1 (Remarks to the Author):

A. Key findings:

The study may inform structural analyses in HRT, as well as therapeutic strategies, with a focus on the emerging avenue of astrocyte-targeted therapies. The authors are to be commended by the hard work and the wealth of tests and animal models. Tests include hippocampal-dependent spatial memory (Morris Water Maze, MWM), working memory (Y-maze) and contextual fear conditioning. Of great value is the use of MRI to dissect out regional changes in association with behavioral deficits. Mouse models include intact and gonadectomized, as well as knock-out mice for estrogen receptor types in astrocytes and neurons.

We appreciate these comments recognizing the breadth of our experiments.

B. Limitations:

B.1. Insufficient clarity.

Conceptual: The gap of knowledge ('understanding brain region-specific, cell-specific and receptor-specific mechanisms') that the authors aim to dissipate should be rephrased. As I understand it, the authors aim to clarify why HRT does not mitigate cognitive deficits and brain atrophy in menopausal women (problem statement) and propose that the conflicting results may arise from paradoxical actions of different estrogen receptors in different cell types, and the fact that readout measures are not directed to the right brain areas. There is a confusion in the said gap between actors (cells and receptors) and effects (brain areas and cognition), for, in the study, brain regions are not specifically targeted. The gene knockout affects cells all over the brain, and clinical therapies are systemically administered. The repeated expression 'region-specific targeting' throughout the text should be eliminated or clarified.

As suggested by the reviewer, we have now removed "gap" from the title as well as throughout the revised manuscript. Also, our phrasing, "Region-specific targeting", warrants clarification. Here, we show functional, pathology, and MRI effects that support future treatments targeting a specific cell and brain region (astrocytes in dorsal hippocampus). Future therapeutics could target this cell and region in midlife females. This does not limit treatments to those affecting only astrocytes in dorsal hippocampus (no other cells and no other regions). That is a pharmacologic bar too high for existing technologies. However, future clinical trials could tailor estrogen type, inclusion criteria, and outcome measures (specific cognitive domains, brain substructure atrophy, PET imaging, pathology of post-mortem or biopsy tissues, etc.) to assess whether a systemic treatment has an effect on astrocytes in dorsal hippocampus in midlife females. This manuscript identifies astrocytes in dorsal hippocampus as such a candidate cell and region.

Experimental: Perhaps it is just me, but the study rationale is not clear, particularly which tests are used, and which ages analyzed. Specifically, in figure 1, in intact mice only MWM is performed at three ages, with negative results, while in Fig 2 (gonadectomized mice) Y Maze and contextual fear conditioning are incorporated, but only in two ages. The comparison of young and midlife mice allows to test the interaction between age and

absence of hormones, but the absence of old mice makes it difficult to conclude whether gonadectomy at midlife accelerates cognitive and structural decline to match that of old ages. Also, Fig. 2a and Fig. 2c repeat data. Perhaps Fig.1 and Fig. 2 should be condensed into a single figure to present the model of intact (normal scenario) and gonadectomized mice (extreme scenario) and conclude which behavioral and structural analyses are more apt to study the effect of hormones. Likewise, in Fig. 4 (studies in ER knockout mice) Y Maze is not used, only midlife mice are tested, and immunohistochemical analyses of GFAP, Iba1 and PSD95 are not performed.

See response to the Editor. Our experimental strategy was focused on midlife females with a set of informative comparators.

Combining former Fig. 1 and 2 (current Fig. 2 and 3) would not align with each question investigated in each figure. An assessment of sex differences is the focus of current Fig. 2, while investigation of midlife females + / - GDX is the focus of current Fig. 3.

As suggested by the reviewer, MWM data are now shown as a merged graph with both between group comparisons of % Time in Target Quadrant and within group comparisons of % Time in Target Quadrant compared to % Time in Other Quadrants. Former Fig. 2a and 2c (new Fig. 3b and 3c) make different comparisons of midlife females, one to midlife males and the other to young females, respectively.

ERbeta knock-outs are shown not only at midlife in former Fig. 4 (new Fig. 5), they are also shown at young age (see Suppl Fig. 1).

As suggested by the reviewer, additional immunohistochemical images and analyses for the ERbeta cKOs are now added to former Fig. 4 (new Fig. 5) of the revised manuscript.

Fig. 3 e,f. I guess that it is normal that '% time in other quadrants' is the mirror/inverse version of '% time in target quadrant'. If so, why is the scale in the Y axis of Fig. 3f different from the one in 3e? For example, a 20% time in target quadrant should correlate with 80% time in other quadrants.

We are not reporting % time in the other quadrant, instead it is % time in the other quadrants plural (an average of the % time in the other 3 quadrants). Thus, % time in other quadrants is not the mirror/inverse version of % time in the target quadrant. For example, 20% time in TQ would align with 80% total time in the other three quadrants, with the average of the three other quadrants = 26.7%.

B.2. Overinterpretation of correlations.

Direct results should be differentiated from interpretations. For example, line 212 'Neuroprotection in females at midlife is mediated by ERbeta in astrocytes' is not a correct label. The experimental design will show the effect of ERbeta removal, but it does not demonstrate that the deleterious mechanisms are the same as those mediating damage caused by gonadectomy, which affects many hormones. Such an equivalence is a correlation. Reversal experiments such as addition of ERbeta agonists to gonadectomized mice may beautifully strengthen the connection the authors wish to make.

As suggested by the reviewer, to beautifully strengthen the connection, "reversal experiments such as addition of ERbeta agonists to gonadectomized mice", are now done (see new Fig. 7 of the revised manuscript).

The statement, “Neuroprotection in females at midlife is mediated by ERbeta in astrocytes” is a fair based on our data. When selective deletion of a receptor (in the cKO) causes a deleterious effect, then the presence of that receptor prevents the deleterious effect (in the WT). Here, cKO and WT were females at midlife.

C.Methodologies:

Supplemental Fig. 1. presents data from estrogen receptor alpha knockout mice that are not described in Methods. These data are important and could be moved to the main text. Likewise, the PCR results confirming successful deletion of estrogen receptor genes should be shown.

As suggested by the reviewer, confirmation of specific deletion of estrogen receptor beta is now shown (see new Fig. 4 of the revised manuscript). Supplemental data on ERalpha knock-out mice are now removed since repeating all outcomes in another knock-out would take focus away from the role ERbeta. Instead we provide additional data on ERbeta including 1) treatment with the ERbeta ligand to reverse hippocampal outcomes, and 2) RNA-sequencing and gene expression analysis of the astrocyte transcriptome in midlife females with deletion of ERbeta in astrocytes.

It would be nice to have images of MRI in different experimental scenarios in Fig. 1.

Other than the substructure delineations overlaid onto the brain image in new Fig. 2 (former Fig. 1), it is unclear what image is being requested. Notably, MRI images from young to midlife to old mice will not appear to differ upon visual inspection. The standard approach required is quantification of neuroanatomic substructure volumes in 3D using atlas-based morphometry, which is what we did.

The quantification of immunohistochemical data is not explained in sufficient detail, including the number and distance between sections, stereological considerations, programs used to quantify, thresholding, background subtraction, etc. Also, images of PSD95 immunostaining, perhaps the extreme cases, would be useful to visually support one of the most interesting finding of the study.

As suggested by the reviewer, the revised manuscript now has detailed descriptions of immunohistochemical analyses, including antibody information and the analysis program used. Representative images of reactive astrocytes, microglia activation, disease-associated microglia, and synapses have each been added (see new Fig. 3 and Fig. 5 of the revised manuscript). Regarding synapses, we have now shown interesting immunostaining and analyses for SYN1 (including images) in the revised manuscript).

D. Style issues:

The authors may consider removing ‘Mind the Gap’ from the title.

As suggested by the reviewer, the title is revised, removing ‘Mind the Gap’.

The manuscript is well written, but there is always room for improvement. In the Abstract ...

The Abstract is rewritten.

Line 52 'during health' is unnecessary. The study does not need to touch on the controversy of whether there is healthy or unhealthy aging, or aging is always unhealthy. Also, the Marxian sentence (from Groucho) 'A better understanding of the effect of aging can provide insights into the effect of aging' should be rephrased for clarity.

We agree with the reviewer that our study does not need to touch on the controversy of healthy or unhealthy aging. Our use of the term disease means neurological disease (not aging in otherwise healthy). The full sentence is: "A better understanding of the effect of brain aging during health can provide insights into the effect of brain aging during disease". "During health" is necessary to contrast with "during disease". This refers two different aging scenarios, whereby mechanisms in one can inform the other.

Line 59. It appears from the sentence that the impact of timing and estrogen type (does it mean receptor type?) has been already clarified according to the literature (refs 6-9), in contradiction with the problem statement, and with line 88 (refs 8, 28). Please, succinctly specify what is known and what are the areas of conflict.

Line 62. This sentence belongs to a review, there is no need to defend the need for separation of sexes in a study using, moreover, a reference from 2012. Likewise, the rest of the paragraphs can be summarized, or moved to the Discussion. For example, it should be highlighted in the Discussion that the mice mirror the sex-dependent evolution of cognition and atrophy in humans.

Lines 212 and 254. These two paragraphs can be condensed into one. Also, the explanation of the contextual fear conditioning (line 256) should be moved to earlier in the text, when the meaning of the different tests is explained.

Wording is changed in the revised manuscript.

Line 91. Again, the study does not inform about brain region-specific therapeutic approaches. What it does is to unravel which brain regions are more affected by the experimental manipulations; nor the authors address the regional heterogeneity of astrocytes or neurons, for their genetic manipulations are global.

See above regarding clarification of "Region-specific targeting".

In Discussion, the clinical implications of the findings can be summarized. Text condensation is recommended.

Condensation is now done in the revised manuscript.

As most articles, including those written by English-speaking authors, the manuscript may benefit from professional copy editing.

Thank you very much for these suggestions which have substantially improved the manuscript!

Reviewer #2 (Remarks to the Author):

Title: While I understand the underlying idea, i.e. that ER β in astrocytes could be an important candidate for future therapeutic targets in cognitive aging in female. I tend to find it overclaiming. The authors do not demonstrate that ER β solely mediates the potential improvement of memorisation in aging, for example using a specific drug targeting those receptors, or losing a positive effect after the deletion.

As suggested by the reviewer, we have now added the use of “a specific drug targeting those receptors”, namely ER beta ligand treatment (see new Fig. 7 of the revised manuscript).

Results using the astrocyte-specific knock-out show that ER beta in astrocytes is necessary to prevent deleterious effects on cognitive outcomes at midlife in females. We do not claim that “ER β solely mediates the potential improvement ...”. Our finding is not mutually exclusive of additional potentially protective effects of other molecules or cells.

1. Behaviour: I was very surprised to read that the authors did not find an age-related defect of spatial memory in the Morris Water Maze (MWM) experiment (Fig1). It is a well-established fact that hippocampus-dependent types of memory, such as spatial, reference, or episodic memory, are sensitive to aging across species (please see the body of work from Catherine Barnes, Howard Eichenbaum, Michela Gallagher, Michael Yassa, Aline Marighetto, Sara Burke, to name a few). In trying to understand this lack of defects, I came to think that the paper would benefit not only from showing the performance during acquisition, at least in the supplementary material, as a control of learning, but also from providing more details about the paradigm. Indeed, depending on the protocol, the memory can go from spatial to procedural (i.e., hippocampus-independent), which remains relatively intact during aging. Alternatively, the authors could also represent the target quadrant exploration during the test as blocks of 15 sec to clarify the mouse behaviour. That being said, I think the high performance from aged mice is due to the fact that the authors tested the mice 2h after the last training session. This could explain the difference between retrieval in the MWM and the fear conditioning, which was tested 24h post-conditioning, and did reveal the age-related memory decline. I would suggest to the authors to homogenise the conditions of behavioural testing throughout the paper, for clarity of understanding and comparison, or at least to address these discrepancies in the discussion. As the age-related decline of hippocampus-dependent memory is such a long-standing fact, I don't think the study benefits from starting with such a clashing result and could deter the readership. Especially since in GDX mice, the defect appears earlier, at “mid-life”, and could be interpreted as early aging. Finally, regarding the behaviour part, I would like to draw the authors' attention to the fact that the dichotomy in the role of the hippocampus to sustain recent but not remote memory has been revised in recent years (Goshen et al., Cell 2011) benefiting from the advancement of optogenetics. While I agree with the authors' conclusion regarding their own result, I would advice to modify the introduction of this part to take into account recent literature rather than Frankland and Bontempi's early work. A minor remark on this part, I am sure what is the authors' rationale for making this data as a supplementary figure only, when this is an interesting result.

We agree with these insightful comments of the reviewer. Various protocol paradigms (timing, etc.) can affect behavioral performance. Here, using the MWM protocol as described, with the sample size as indicated, we did not observe a significant defect in spatial reference memory in the Morris Water Maze in the 2hr delay probe trial in gonadally intact, healthy females or males up to the ages tested in the environment of the UCLA Behavioral Core Facility. This lack of a significant deficit has been seen before in healthy control groups of studies focusing on AD mice where healthy mice served as

the cognitively normal control, albeit often tested at ages a few months earlier. In agreement with the reviewer, we observed that older ages had a learning speed difference (older slower than younger), so a training learning graph could be added as a supplemental figure upon request. However, our focus here is not to do additional behavioral tests to show significant deficits in gonadally intact mice using a given paradigm. Instead, our focus is on the significant worsening observed in GDX females compared to gonadally intact females using a standard MWM protocol.

2. ER β cKO. I think it would be helpful to provide more details regarding the origin of the mice and selectivity of the conditional KO.

As suggested by the reviewer, this is now done (see new Fig. 4 in the revised manuscript).

3. Neurodegeneration: One major point that I would like to draw the authors' attention towards, is their multiple reference to neurodegeneration throughout the paper, and the lack of distinction between normal aging and pathological aging. From what I understand, the authors seem to suggest that their study addresses neurodegeneration. However, the study has been conducting in C57Bl/6j mice, which do not develop pathological aging (i.e., dementia, Alzheimer's-like memory defects), as demonstrated in my previous point, these wildtype aged mice can demonstrate high performance in certain conditions (Fig1), which would not be the case in pathological aging. In normal aging, the memory defects are associated with atrophy, but not neuronal loss, unlike pathological aging. Moreover, the underlying mechanisms, and progression of alterations are widely distinct according to the type of aging. Therefore, I do not think it is proper in their condition to refer to neurodegeneration and would reframe the paper as normal aging. They could address the potential importance of their work regarding pathological aging in the discussion, but the use of wildtype mice do not support the direct extrapolation towards neurodegeneration.

It is unclear what is meant by the need to "reframe the paper as normal aging". As the reviewer points out: 1) Our study was "conducted in C57Bl/6j mice, which do not develop pathological aging (i.e., dementia, Alzheimer's-like memory defects)", and 2) "these wildtype aged mice can demonstrate high performance in certain conditions". Our manuscript is framed on the mice we used, those with "normal aging", and these two correct statements by the reviewer show that we have conveyed that.

We did not make a "distinction between normal aging and pathological aging". As stated by Rev #1, we are not entering "the controversy of whether there is healthy or unhealthy aging, or aging is always unhealthy",

Use of the word "neurodegeneration" is not problematic when describing a condition characterized by cognitive decline, glial activation, synaptic loss, and brain atrophy, indeed what occurs in midlife females either GDX or with selective deletion of ER beta in astrocytes.

4. Discussion: I think the work would benefit from stressing out throughout the text and in particular the discussion, the novelty of the research, and replacing the work within the literature. It is my belief that aside from the very interesting distinctive role of ER β in astrocytes vs. neurons, the rest of the work has in some form been already reported. Age-related memory deficits (see authors in part 1 for reference); There is also a large body of work on the sexual dimorphism in cognitive aging (e.g. Frick et al., 2000; for review, Frick et al 2008), and on the role of estrogens in the brain and learning and memory, and effect of ovariectomy, which deserve to be cited in my opinion (L. Galea, K. Frick, E. Waters, M. Adams, C. Wolley, B.

McEwen – non-exhaustive list). Similarly, the atrophy of the hippocampus has been demonstrated multiple times (for review Barnes and Burke, 2010). Finally, the reduction in spine density following ovariectomy has been previously reported by the McEwen laboratory (Gould et al., 1990). I do not intend to denigrate the study; I simply believe that the authors would benefit from valorising their findings in light of the existing literature.

Final point on the discussion, I fail to understand why the authors address hormone replacement and MS in such length, when their study does not use estrogen supplementation, potential effect on alteration similar to MS, and that MS have limited cognitive effect. Rather, I would be very interested in reading the authors' take on the integration of their findings about sexual dimorphism, from cognition to synaptic loss and the specific role of glial cells in these processes.

The revised manuscript now includes estrogen supplementation (see new Fig. 7). We thank the reviewer for highlighting our finding of a “very interesting distinctive role of ER β in astrocytes vs. neurons”. We have now further added additional novel findings on gene expression analyses of astrocyte transcriptomes in midlife females with ER beta selectively deleted. This revealed an exciting link to previous work by others on the role of glucose metabolism in brain during menopause, albeit previously not studied in hippocampal astrocytes.

The literature cited by this knowledgeable reviewer is excellent and could be the basis for a stand-alone review article. We agree with the reviewer that some of our work has been done in parts in papers by others. The strength here is integration of a breadth of complementary studies (some done before, some not) in the same paper, namely behavioral studies, *in vivo* MRI for regional atrophy, underlying neuropathology, cell-specific knock outs, RNA-sequencing, and estrogen receptor specific treatment reversal). Rather than reviewing parts published in the past, we took our discussion in a future direction. We discussed implications of the totality of our findings on clinical translation. This addresses the unmet need for a hormone replacement therapy designed more specifically and based on neuroscience mechanisms underlying cognitive outcomes in midlife females.

Regarding justification of the discussion of MS, these patients are predominantly female, have substantial cognitive disability, and menopause confers overall disability worsening. In addition, clinical trials testing a naturally occurring ER beta ligand (estriol) treatment showed neuroprotective effects in women with MS. This published work warrants discussion of the potential for repurposing this treatment approach to cognitive deficits in healthy women with menopause.

We thank the reviewer very much for their interest in getting the “authors' take on the integration of their findings about sexual dimorphism, from cognition to synaptic loss and the specific role of glial cells in these processes”. Dr. Voskuhl is an expert on the study of sex differences, and this was the starting point of this manuscript (new Fig. 1 and 2). However, the trajectory of this manuscript is to use initial sex differences findings as a basis for subsequent research focused on gaining cellular and molecular insights relevant to midlife females with cognitive deficits. A discussion of sexual dimorphism as described by the reviewer would align better with a manuscript that studied both females and males with the full breadth of our outcomes from young to midlife to old ages. While that is interesting, it would be a different paper.

Minor comment: I think the reader would benefit from the homogenisation of the statistical report, which at the moment is alternatively in the legend or the text.

As the reviewer suggested, this is now done in the revised manuscript. We also have a statistical summary ready for submission pending the response to this appeal to proceed.

Looking forward to reading your response,
AS Al Abed

Thank you for your insightful comments.

Reviewer #3 (Remarks to the Author):

1) The authors use GFAP-cre mice to constitutively delete ER beta in astrocytes. These mice cannot be used to conclude about implication of astroglial proteins, when these proteins are also expressed in neurons.

There are indeed major caveats in using the GFAP-Cre mice, as it has been well documented by numerous groups that deletion occurs in both neurons and astrocytes, since GFAP is expressed in precursor cells during early development. This confounds the authors major conclusion that deletion of ER beta from astrocytes contributes to cognitive deficits and brain atrophy in menopausal female.

A few papers here listed well describe these caveats :

"Looks Can Be Deceiving: Reconsidering the Evidence for Gliotransmission"

<https://doi.org/10.1016/j.neuron.2014.12.003>

Germ-Line Recombination Activity of the Widely Used hGFAP-Cre and Nestin-Cre Transgenes
<https://www.ncbi.nlm.nih.gov/pmc/articles/PMC3857304/>

Expression Specificity of GFAP Transgenes

<https://link.springer.com/article/10.1007/s11064-004-6881-1>

Other experiments should thus be combined with the use of these mice, such as the use of inducible cKO mice and/or rescue experiments targeted to astrocytes (re-expression of ER beta specifically in astrocytes in the astroglial cKO mice).

There are different subtypes of *mGFAP-Cre* lines. The *mGFAP-Cre* line 77.6 used in this manuscript has now been characterized, and specificity for astrocytes and not other cell types (including neurons) was shown (see new Fig. 4 in the revised manuscript).

Recent publications used the *mGFAP-Cre* 77.6 line for selective gene deletion in astrocytes: 1) Deerhake ME et al, Immunity, 2021, and 2) Jia YF et al, Behav. Brain Res. 2021. They also showed specificity of deletion in astrocytes.

The reviewer states that *GFAP-Cre* driven deletion in our experiments is not specific to astrocytes, that ERbeta was also deleted from neurons, so our effects on cognitive outcomes at midlife may be due to ERbeta deletion in neurons (not astrocytes). If that were true, then one would see effects on cognitive outcomes in the neuronal ERbeta cKO (*NSE-Cre*), but that was not the case (see former Fig 4, new Fig. 5).

Specificity of *GFAP-Cre* driven selective gene deletion can differ not only between different *mGFAP-Cre* lines, but also between *mGFAP-Cre* versus *hGFAP-Cre*, and between sexes. The *mGFAP-Cre* 77.6 line has shown some expression of Cre recombinase in the male, but not the female, germline

<https://www.jax.org/strain/024098>). To avoid this ectopic recombination, the *mGFAP-Cre* 77.6 allele should not be in male parents in the breeding scheme. In our experiments, *mGFAP-Cre* 77.6: ER β^{ff} females were crossed with ER β^{ff} males (without the *mGFAP-Cre* 77.6 allele), thereby avoiding germline ectopic recombination. This breeding instruction about the *mGFAP-Cre* 77.6 line is available from Jackson Labs, but could be added to our Methods section upon request. We can also clarify the sex of the parents in our breeding strategy in the Methods section if needed.

Regarding use of an inducible Cre mouse line, cognitive and hippocampal outcomes observed at midlife were not observed in astrocyte ERbeta cKO mice at age 3 months (see Suppl. Fig. 1). Thus, cognitive and hippocampal outcomes observed in astrocyte ERbeta cKO mice at midlife were not due to effects on brain development.

2) There is no validation of these newly generated astroglial- and neuronal-specific ER beta transgenic models. One would like to see quantification of ER beta expression in astrocytes and neurons in both cKO mice (*gfap-cre:ER betafl/fl* and *nestin-cre:ER betafl/fl*).

As suggested by the reviewer, validation of ERbeta deletion specifically in astrocytes is now done (see new Fig. 4 of the revised manuscript).

Validation of ERbeta deletion specifically in neurons using the *NSE-Cre* has been validated and published, see Spence, et al, J. Neuroscience, 2013 (as cited in this manuscript).

3) There is a lack of details regarding the mice (origin (stock number or laboratory source), breeding strategy controlling germline recombination, appropriate littermate controls. This precludes evaluation of whether the authors have used appropriate controls.

As suggested by the reviewer, details of information about mice, including references, have now been added to the Methods section. Also, schematics of breeding have been added to new Fig. 4a and 4d of the revised manuscript.

4) The authors use GFAP immunostaining to assess astroglial reactivity. GFAP expression can vary independently of astroglial reactivity. The authors should thus use additional markers (such as vimentin, stat3, GS...) and analyse astrocyte morphology to conclude about astrocyte reactivity.

As suggested by the reviewer, reactive astrocytes are now stained as double positive: LCN2⁺GFAP⁺ (see Fig. 3, Fig. 5, and Fig. 7 of the revised manuscript).

5) The authors report synapse loss in menopausal female. They however only performed PSD95 staining to assess synapses. A synapse is composed of a pre- and a postsynaptic element and can only be identified by the colocalization of both elements. Staining for both pre- and postsynaptic markers should thus be performed to quantify synapse number.

Pre-synaptic staining for SYN1 is done in the revised manuscript (see Fig. 3, Fig. 5, and Fig. 7)

6) This manuscript completely lacks mechanistic insights. How ER beta in astrocytes contributes to brain structure and cognitive performance in male and female? Does this require a crosstalk with other cell types such as microglia?

Mechanistic insights are shown by a series of causality experiments. 1) An ovarian hormone by age interaction was found, whereby both midlife aging and loss of ovarian hormones caused cognitive decline, neuropathology, and atrophy of dorsal hippocampus by *in vivo* MRI. 2) Selective deletion of estrogen receptor beta (ER β) in astrocytes, but not neurons, caused cognitive decline, neuropathology, and atrophy of dorsal hippocampus by *in vivo* MRI. 3) Complementing our two loss-of-function causality experiments (gonadectomy and specific deletion of ER β in astrocytes), gain-of-function experiments using ER β ligand treatment of midlife females showed that treatment caused reversal of outcomes. 4) Gene expression differences in hippocampal astrocyte transcriptomes from midlife females with selective deletion of ER β vs WT in astrocytes revealed Gluconeogenesis I and Glycolysis I pathways as the most differentially expressed pathways. Further studies of a key gene in the Gluconeogenesis I pathway, *Eno1*, were also done (see new Fig. 6 in the revised manuscript).

Additional mechanisms related to cross talk between astrocytes and microglia are interesting and discussed. Further experiments addressing this possibility are beyond scope here.

Reviewer #4 (Remarks to the Author):

The narrative is very interesting, the results important and well contextualized. The work is extremely significant and timely. The methodology is rigorous, although a few important pieces of evidence are missing to have a more balanced picture and close the story (detailed below). I also have doubts on some of the statistical approaches chosen. The paper is easy to follow and very well written.

There are a few aspects that can be improved. Below, my suggestions.

Thank you!

1

Fig.1 : the information contained in a-b and c-d is redundant. In addition, doing twice a scatter plot, which distributes points random, on the same data generates plots which are visually different, which is awkward. The same applies to Fig. 2 a-b and c-d, and Fig. 4 a-b. I think half of the plots can safely go to supplementary.

As suggested by the reviewer, to remove redundancy MWM data are now shown as a merged graph with both between group comparisons of % Time in Target Quadrant and within group comparisons of % Time in Target Quadrant compared to % Time in Other Quadrants. This is done in former Fig. 2 (new Fig. 3) and in former Fig. 4 (new Fig. 5)

2

Following the previous comment, I think the most correct statistical approach would be a single ANOVA for the water maze data where sex is a factor (in addition to age and quadrant), followed by appropriate post-hoc comparisons. The same applies for MRI volumetric data, where ROI should be another factor (cortex, striatum, HC).

In the revised manuscript, given the relatively small sample size and unequal variance, MWM was analyzed using a nonparametric test, Mann Whitney *U* test. All volumetric MRI data were analyzed using a two-way ANOVA (age, sex) and then followed up with two-tailed Welch's t-tests, appropriate for these comparisons. See item 4 below.

3

In Fig 1 h, it seems that there is no volume change in HC in male mice across the lifespan. This results somehow weakens the point made by the authors in the rest of the paper, about the importance for neuroprotection across the lifespan of the ERb astrocyte receptors specifically in the dorsal hippocampus, unless the dHC is only important in ageing female, which is hard to digest, unless supported by evidence. Generally speaking, the paper is missing showing and discussing in males at least some of the analyses performed in females, for example Fig 2 g-i.

The reviewer is correct. While females had dorsal hippocampal atrophy from midlife to old age, males did not.

Analysis of data from the Genotype-Tissue Expression (GTEx) project examined sex differences in gene expression across 44 tissues in humans and showed that 37% of all genes exhibit sex-biased expression in at least one tissue (Oliva et al., Science, 2020). In another study using the same dataset focusing on 29 human healthy tissues, whole-genome expression profiles showed distinct sex-biased regulatory networks in each tissue (Lopes-Ramos, Cell Rep. 2020). Finally, sex differences in gene expression are region-specific and cell-specific within the brain (Kim-Hellmuth, et al., Science, 2020). These studies underscore the pervasiveness and complexity of sex differences in gene expression during health, which can be distinct depending on the brain region and cell involved.

Sex differences in substructure volumes and rates of atrophy have previously been shown in MRI studies in humans and mice. Dorsal hippocampus and ventral hippocampus differ in regard to which is larger in females versus males, each when normalized for brain size.

Given the above evidence in the sex differences field, we did not find it particularly surprising that dorsal hippocampus may undergo atrophy with aging in females and not males. That said, this manuscript did not pursue why dorsal atrophy did not occur in males with aging. A related question is why did males have gradual atrophy with aging in frontal cortex and striatum, but not in dorsal hippocampus? It could be due to direct effects of testosterone or dihydrotestosterone acting on androgen receptors in hippocampus in males. This is only one of several possibilities. Space limitations do not permit our speculations about males during aging since our studies focused on females during aging.

4

Fig 1, supplementary excel table: why using Welch test? Is it because the samples have unequal variance? Please specify in the methods.

Yes, the samples do exhibit unequal variances, as is the case in almost all biological measures. Thus, the use of Welch's t-test is appropriate here since it performs better than the Student's t-test when sample sizes and variances are unequal between groups, and it gives identical results when sample sizes and variances are equal. We routinely use Welch's t-test for volumetric analyses in atlas-based morphometry.

5

Fig 1: It is noteworthy that from Fig. 1 there is no relation between structure and function, at

least in the context of ageing. If it is a matter of different sensitivities of the two techniques employed (MRI volumetry vs behaviour), as hinted by the authors, I suggest they discuss the issue and report the power analysis used for determining the sample size. Perhaps the behavior is underpowered?

See response to Reviewer #2 regarding not observing a significant defect in spatial reference memory in the Morris Water Maze in the 2hr delay probe trial in gonadally intact, healthy females or males up to the ages tested in the conditions of the UCLA Behavioral Core Facility given the sample size used. Sample sizes were driven by the number of mice needed to generate a robust minimum deformation atlas (MDA) for atlas-based morphometry (MacKenzie-Graham NeuroImage 2012). Logistically, with several groups (various ages, two sexes, and/or GDX vs sham surgery, cKOs vs WT, treatments) and several outcomes, sample size was also limited by what was practically feasible by personnel and costs. Data from *in vivo* MRI is expected to be more sensitive than clinical testing in mice. This aligns with *in vivo* MRI for brain atrophy being a more sensitive outcome than clinical outcomes in humans. Indeed, MRI is very frequently used as a more sensitive biomarker in Phase 2 clinical trials prior to designing Phase 3 trials which require much larger sample sizes since they have a clinical primary outcome (such as cognitive performance).

6

Fig.2: Similarly to comments 1 and 2, the figures of the water maze are redundant, and the statistical model could be just one for each test but include everything (gender, therapy, life stage).

As suggested by the reviewer, redundancy of MWM data has been decreased by making a single merged panel for between groups and within groups comparisons. See comments above regarding statistical tests used.

7

Fig. 2 f: why freezing was only measured at midlife?

Fear conditioning was done to expand upon MWM test performance at midlife in females, the group and age of focus. It was not done at all ages, in both sexes, in + / - GDX, or in cKOs. Therefore, fear conditioning has now been removed from the revised manuscript.

8

Fig. 2 j-k: I suggest to only plot regression line when significant. Also, there seems to be a lot of points in the plots. Are all animals (both sexes, all ages, GDX/sham) pulled together? If so, which is the rationale? Perhaps the different groups could be shown in the plot by coloring the datapoints differently.

As suggested by the reviewer, dot plots in former Fig. 2 (new Fig. 3 p,q) have now been made using distinct coloring of datapoints (see below). This can replace the panel with all black dots upon request. The rationale of pooling groups is to increase the sample size when asking if cognitive performance (% Time in Target Quadrant) correlates with dorsal (p) or ventral (q) hippocampal volume by *in vivo* MRI. The plots nicely show a positive correlation between better cognitive performance and higher volumes of dorsal (but not ventral) hippocampus. This is presented in our manuscript since *in vivo* MRI for hippocampal atrophy is not usually done in papers showing behavioral testing in mice.

Possible panel replacement in new Fig. 3. % time in TQ correlated with (p) dorsal hippocampus volume ($r=0.28$; $p=0.011$), not with (q) ventral hippocampus volume ($r=0.027$; $p=0.809$). Dot colors: black (astrocyte ERβ cKO, old), yellow (astrocyte ERβ cKO, young), blue (neuron ERβ cKO, old), green (neuron ERβ cKO, young), red (WT, old), pink (WT, young).

9

Fig. 3, Neuropathology: the authors talk about “area fraction” but what is exactly measured and how? Can distinct aspect of the morphology be measured (ramifications density, cell body size, cell number etc.? This is important as the term “glia activation” can refer to distinct morphological changes happening to the cell. In general, the method->histology section is missing details needed for the reader to be able to reproduce the analyses.

The term “area fraction” is widely used during double-labelling. For example, positive labeling in a specific cell type was quantified and graphed as the percentage labeling of LCN2+ area within the GFAP+ area. A level of detail needed for the reader to reproduce histology (information about antibodies, dilution factors) is now in the Methods section of the revised manuscript.

10

Line 194: the notation is misleading. Perhaps by “direct” and “indirect” correlations the authors mean linear positive and linear negative? Or do the authors really mean indirect as mediated by something else? If so, specify.

The reviewer is correct. Direct is positive and indirect is negative.

11

Fig 3 is missing the plots of the correlations between time in TQ and gliosis (data is discussed but not shown).

The correlation between % Time in Target Quadrant and glial activation markers are stated in the results section. Plots showing the correlations between % Time in Target Quadrant and LCN2+GFAP+ astrocytes, as well as % Time in Target Quadrant and MHCII+ IBA1+ microglia are presented below. This can be added as additional supplemental figure upon request.

Possible additional supplemental figure. Correlation of MWM cognitive performance with glia activation. a-b) % Time in target quadrant is negatively correlated with a) LNC2⁺GFAP⁺ astrocytes ($r=-0.46203$, $p=0.017$) and b) MHCII⁺ IBA1⁺ microglia ($r=-0.40723$, $p=0.048$). red dot, Young Sham; red dot with black rim, Young GDX; black square, Midlife Sham; white square, Midlife GDX females. Pearson correlation analyses.

12

Following the rationale of the paper, Fig 4 is missing the neuropathology. How are the astrocytes affected by ERb deletion? Are they more activated at midlife?

As suggested by the reviewer, this neuropathology has now been done (see new Fig. 5e-l in the revised manuscript).

13

Following the previous comment but more generally, what is ultimately the mechanism that the authors identified as leading to atrophy and worst cognition in ageing females? Is it astrogliosis? If so, the authors should report more neuropathological data (or other equivalent astrogliosis markers) to make a stronger point.

In addition to more neuropathology, we have also added new Fig. 6 showing gene expression in astrocytes by RNA-sequencing. See response #6 to Reviewer #3.

14

It is interesting that there is no cognitive effect in young female mice which lacks endogenous hormones or receptors. What about HC volume?

There were no differences in HC volumes in young female GDX vs young female sham. See the figure below, which can be added as a supplemental figure upon request. This, together with our other findings, reveals an ovarian sex hormone by age interaction. In essence, it is deleterious to have BOTH aging to midlife AND to lose ovarian hormones.

Possible additional supplement figure. Gonadectomy did not induce hippocampal atrophy at young age. Substructure volumes, assessed by MRI, taken as a percentage of intercranial volume (ICV) are shown for **(a)** hippocampus, **(b)** dorsal hippocampus, and **(c)** ventral hippocampus. There were no differences in substructure volumes between groups at young age. n=8 for each group.

15

As the authors point out in the discussion, microglia also have ERb. Can the author show and compare some histology of microglia in animals lacking endogenous receptors and animals lacking ERb in astrocytes?

As suggested by the reviewer, this is now added to the revised manuscript. See double-labelling stains for MHCII & IBA1 as well as for CLEC7A & P2RY12. This is done in sham vs GDX (new Fig. 3), cKO vs WT (new Fig. 5), and ER beta ligand treatment (new Fig. 7).

REVIEWER COMMENTS

Reviewer #3 (Remarks to the Author):

The authors have addressed several of my comments in the revised manuscript. Yet there are still a few points that were not satisfactorily addressed and needs to be done:

1) "To avoid this ectopic recombination, the mGFAP-Cre 77.6 allele should not be in male parents in the breeding scheme. In our experiments, mGFAP-Cre 77.6: ER β /f females were crossed with ER β /f males (without the mGFAP-Cre 77.6 allele), thereby avoiding germline ectopic recombination. This breeding instruction about the mGFAP-Cre 77.6 line is available from Jackson Labs, but could be added to our Methods section upon request. We can also clarify the sex of the parents in our breeding strategy in the Methods section if needed."

Please add in the methods section the breeding strategy.

2) "As suggested by the reviewer, validation of ERbeta deletion specifically in astrocytes is now done (see new Fig. 4 of the revised manuscript). Validation of ERbeta deletion specifically in neurons using the NSE-Cre has been validated and published, see Spence, et al, J. Neuroscience, 2013 (as cited in this manuscript)."

To be fully convinced by the astroglial cKO mice in, the authors need to illustrate that the expression of ER beta is intact in neurons from the gfap-cre:ER betafl/fl.

3) "Pre-synaptic staining for SYN1 is done in the revised manuscript (see Fig. 3, Fig. 5, and Fig. 7)"

I understand the SYN1 staining is done and illustrated in these figures, but it is not used in the manuscript to quantify synapse number by counting the number of colocalized SYN1 and PSD95 puncta. This needs to be done.

Reviewer #4 (Remarks to the Author):

The authors have successfully addressed my concerns and those from other reviewers. Minor comments:

Possible panel replacement in new Fig. 3: I think that the plot can stay b/w.

"Direct" and "indirect" to qualify correlation is used but not precise. Indirect strictly speaking means "mediated by". I recommend using positive or negative.

Plots showing the correlations between % Time in Target Quadrant and LNC2+GFAP+ astrocytes, as well as % Time in Target Quadrant and MHCII+ IBA1+ microglia: I would add them in the supplementary.

Possible additional supplement figure. Gonadectomy did not induce hippocampal atrophy at young age. Also suitable for supplementary material.

REVIEWER COMMENTS followed by **Point-by-point Responses.**

Reviewer #3 (Remarks to the Author):

Comment 1) "To avoid this ectopic recombination, the mGFAP-Cre 77.6 allele should not be in male parents in the breeding scheme. In our experiments, mGFAP-Cre 77.6: ER β ^{f/f} females were crossed with ER β ^{f/f} males (without the mGFAP-Cre 77.6 allele), thereby avoiding germline ectopic recombination. This breeding instruction about the mGFAP-Cre 77.6 line is available from Jackson Labs, but could be added to our Methods section upon request. We can also clarify the sex of the parents in our breeding strategy in the Methods section if needed." Please add in the methods section the breeding strategy.

Response 1) Now added to the methods section as well as referred to in the results section of the revised manuscript is the following:

In breeding of our mouse lines, Cre recombinase alleles were always inherited from females (mother), not from males (father). This is because one Cre line showed ectopic expression of Cre recombinase in the male germline (mGFAP-Cre 77.6 line; <https://www.jax.org/strain/024098>). Thus, paternal inheritance of Cre recombinase should be avoided. To this end, our breeding pairs were: a) GFAP-Cre:ER β ^{f/f} females with ER β ^{f/f} males (to generate astrocyte ER β cKO mice), b) NSE-Cre:ER β ^{f/f} females with ER β ^{f/f} males (to generate neuron ER β cKO mice), c) GFAP-Cre:RiboTag females with RiboTag males (to generate astrocyte RiboTag mice), and d) GFAP-Cre:RiboTag:ER β ^{f/f} females with RiboTag: ER β ^{f/f} males (to generate astrocyte ER β cKO RiboTag mice).

Labels of Male or Female to parents during breeding have also been added to Fig. 4 of the revised manuscript.

Comment 2) "As suggested by the reviewer, validation of ERbeta deletion specifically in astrocytes is now done (see new Fig. 4 of the revised manuscript). Validation of ERbeta deletion specifically in neurons using the NSE-Cre has been validated and published, see Spence, et al, J. Neuroscience, 2013 (as cited in this manuscript)." To be fully convinced by the astroglial cKO mice in, the authors need to illustrate that the expression of ER beta is intact in neurons from the gfap-cre:ER beta^{f/f}.

Response 2) In the revised manuscript a supplemental figure is added focusing on ER β expression in neurons. ER β expression was intact in neurons of the astrocyte-ER β cKO (see Supplemental Fig. 3).

Comment 3) "Pre-synaptic staining for SYN1 is done in the revised manuscript (see Fig. 3, Fig. 5, and Fig. 7)". I understand the SYN1 staining is done and illustrated in these figures, but it is not used in the manuscript to quantify synapse number by counting the number of colocalized SYN1 and PSD95 puncta. This needs to be done.

Response 3) Co-localization of pre-synaptic staining for SYN1 and post-synaptic staining for PSD95 is now done in the revised manuscript (see Fig. 3h, Fig. 5l, and Fig. 7f).

Reviewer #4 (Remarks to the Author):

Comment 1) The authors have successfully addressed my concerns and those from other reviewers.

Response 1) Thank you.

Comment 2) Minor comments:

Possible panel replacement in Fig. 3: I think that the plot can stay b/w.

Response 2) As recommended by the reviewer, panels (in former Fig. 3p,q; current Fig. 2m,n) showing correlation analyses between Dorsal Hippocampus volume and % Time in Target Quadrant or between Ventral Hippocampus volume and % Time in Target Quadrant will stay black and white.

Comment 3) "Direct" and "indirect" to qualify correlation is used but not precise. Indirect strictly speaking means "mediated by". I recommend using positive or negative.

Response 3) The word "direct" correlation has now been revised to "positive" correlation to describe results in Fig. 2m. The word "indirect" correlation has now been revised to "negative" correlation to describe results in Supplemental Fig. 2.

Comment 4) Plots showing the correlations between % Time in Target Quadrant and LNC2+GFAP+

astrocytes, as well as % Time in Target Quadrant and MHCII+ IBA1+ microglia: I would add them in the supplementary.

Response 4) These two plots showing correlations have now been added to supplementary materials (Supplemental Fig. 2).

Comment 5) Possible additional supplement figure. Gonadectomy did not induce hippocampal atrophy at young age. Also suitable for supplementary material.

Response 5) This figure has now been added to supplementary materials (Supplemental Fig. 1).